# Provably Cost-Sensitive Adversarial Defense via Randomized Smoothing

Yuan Xin [1]   Dingfan Chen [2]   Michael Backes [1]   Xiao Zhang [1]

## Abstract

As ML models are increasingly deployed in critical applications, robustness against adversarial perturbations is crucial. While numerous defenses have been proposed to counter such attacks, they typically assume that all adversarial transformations are equally important, an assumption that rarely aligns with real-world applications. To address this, we study the problem of robust learning against adversarial perturbations under cost-sensitive scenarios, where the potential harm of different types of misclassifications is encoded in a cost matrix. Our solution introduces a provably robust learning algorithm to certify and optimize for cost-sensitive robustness, building on the scalable certification framework of randomized smoothing. Specifically, we formalize the definition of cost-sensitive certified radius and propose our novel adaptation of the standard certification algorithm to generate tight robustness certificates tailored to any cost matrix. In addition, we design a robust training method that improves certified cost-sensitive robustness without compromising model accuracy. Extensive experiments on benchmark datasets, including challenging ones unsolvable by existing methods, demonstrate the effectiveness of our certification algorithm and training method across various cost-sensitive scenarios.

## 1. Introduction

Recent studies have revealed that deep neural networks (DNNs) are highly vulnerable to classifying adversarial examples (Biggio et al., 2013; Szegedy et al., 2013; Goodfellow et al., 2014), highlighting a critical weakness in these models that poses significant risks for safety-critical ap-

plications such as autonomous driving, financial systems, and healthcare diagnostics. In response, numerous defenses have been proposed to improve model robustness, which primarily fall into two categories: *empirical defenses* (Goodfellow et al., 2014; Gu & Rigazio, 2014; Lyu et al., 2015; Papernot et al., 2016; Kurakin et al., 2016; Madry et al., 2017; Zhang et al., 2019; Carmon et al., 2019) and *certifiable methods* (Raghunathan et al., 2018; Wong & Kolter, 2018; Gowal et al., 2018; Cohen et al., 2019; Lecuyer et al., 2019; Jia et al., 2019; Li et al., 2019).

Empirical defenses, such as adversarial training (Goodfellow et al., 2014; Madry et al., 2017; Carmon et al., 2019), defensive distillation (Papernot et al., 2016), and gradient masking (Gu & Rigazio, 2014; Lyu et al., 2015), typically propose to enhance robustness by explicitly incorporating adversarial examples into the model's training process or by modifying the training algorithm to make the model less sensitive to input perturbations. These methods are evaluated based on their performance against known attacks and can be efficiently deployed in practice. However, they are engaged in an everlasting arms race, as new adaptive attacks are continually developed that can easily break these empirical defenses (Carlini & Wagner, 2017; Athalye et al., 2018; Uesato et al., 2018; Croce et al., 2022). In contrast, certifiable methods provide robustness guarantees against any perturbation within the constraint set, thereby avoiding the risks of being compromised by new attacks. Specifically, certifiable methods (Raghunathan et al., 2018; Wong & Kolter, 2018; Gowal et al., 2018; Cohen et al., 2019; Lecuyer et al., 2019; Jia et al., 2019; Li et al., 2019) produce a robustness certificate that assures the model's prediction will remain unchanged within a specified norm-bounded perturbation ball around any test input. These methods then train models to be provably robust with respect to the derived certificate, offering a stronger and more reliable form of robustness compared to empirical defenses.

Most existing defenses are designed to improve overall model robustness, assuming the same penalty on all kinds of adversarial misclassifications. For real-world applications, however, it is likely that certain types of misclassifications are more consequential than others (Domingos, 1999; Elkan, 2001). For instance, misclassifying a malignant tumor as benign in the application of medical diagnosis is much more detrimental to a patient than the reverse. Therefore, instead

---

[1]CISPA Helmholtz Center for Information Security, Saarbrücken, Germany [2]Max Planck Institute for Intelligent Systems, Tübingen, Germany. Correspondence to: Xiao Zhang <xiao.zhang@cispa.de>.

*Proceedings of the $42^{nd}$ International Conference on Machine Learning*, Vancouver, Canada. PMLR 267, 2025. Copyright 2025 by the author(s).

of solely focusing on overall robustness, the development of defenses should account for the difference in costs induced by different adversarial examples. In line with prior literature on cost-sensitive robust learning (Domingos, 1999; Asif et al., 2015; Zhang & Evans, 2019; Chen et al., 2021), we aim to develop models that are robust to cost-sensitive adversarial misclassifications, while maintaining the standard overall classification accuracy. Nevertheless, previous methods for promoting cost-sensitive robustness (Domingos, 1999; Asif et al., 2015; Chen et al., 2021; Zhang & Evans, 2019) are either hindered by their foundational reliance on heuristics, which often fall short of providing a robustness guarantee or suffer from inherent scalability issues when certifying robustness for large models and perturbations.

To achieve the best of both worlds, we propose to learn provably cost-sensitive robust classifiers by leveraging randomized smoothing (Liu et al., 2018; Cohen et al., 2019; Salman et al., 2019), an emerging robustness certification framework known for its broad applicability and scalability. However, adapting the standard randomized smoothing algorithm to certify cost-sensitive robustness presents unique challenges. In addition, optimizing smoothed classifiers for cost-sensitive robustness proves more complex than standard cost-sensitive learning due to the added complexity introduced by the smoothing operator and the random Gaussian sampling process. The varying structures of cost matrices further call for a flexible, targeted optimization scheme that moves beyond optimizing solely for overall robustness.

**Contributions.** To the best of our knowledge, we are the first to provide a scalable, robust certification and training algorithm for cost-sensitive robustness that is applicable to challenging high-dimensional data distributions (e.g., medical images) and large models (e.g., deep ResNet). Our key contributions are summarized as follows:

- We introduce the notion of *cost-sensitive certified radius* (Definition 4.1) and prove that our definition guarantees a larger certified radius for any cost-sensitive scenario compared to the standard certified radius (Theorem 4.2).

- We develop an easy-to-implement certification algorithm based on Monte Carlo sampling (Algorithm 1), ensuring a tight certificate for cost-sensitive robustness (Theorem 4.4). Additionally, we propose adaptive training methods for cost-sensitive robustness, ranging from standard reweighting techniques to advanced strategies that optimize the certified radius for smoothed classifiers across data subgroups (Section 5), effectively expanding the robust radius compared to non-optimized approaches.

- Comprehensive experiments across various benchmark and real-world medical datasets demonstrate that our margin-based approach significantly outperforms baselines in certified cost-sensitive robustness across diverse

training configurations and schemes, while largely maintaining overall standard accuracy (Section 6).

## 2. Related Work

**Certifiable Defenses.** Certifiable defenses aim to formally guarantee the robustness of a classifier, ensuring that its output remains consistent within a neighboring region around any input, usually defined by some $\ell_p$-norm distance metric. These methods can be divided into two main categories: *complete* and *conservative* (i.e., "sound but incomplete"). *Complete methods* (Pulina & Tacchella, 2010; Huang et al., 2017; Katz et al., 2017; Tjeng et al., 2019; Wong & Kolter, 2018; Wong et al., 2018; Raghunathan et al., 2018; Dvijotham et al., 2018) strive to exactly determine whether any norm-bounded perturbation can cause the classifier to alter its prediction for any given input. A representative conservative certification framework is *Randomized smoothing* (Cohen et al., 2019), which offers a scalable alternative by transforming any classifier into a smoothed version with probabilistic robustness guarantees, ensuring stable predictions within an $\ell_2$-norm ball around any input. This flexible approach is well-suited for deep networks and large datasets, addressing the scalability limitations of complete methods. Building on the randomized smoothing framework, several training techniques have been developed to enhance certifiable robustness, including applying adversarial training (Salman et al., 2019) or denoising diffusion models to improve the base classifier (Carlini et al., 2022; Xiao et al., 2022; Zhang et al., 2023), and direct optimization of the certified radius (Zhai et al., 2020). However, none of the aforementioned works target optimization for cost-sensitive adversarial defense.

**Cost-Sensitive Learning.** Cost-sensitive learning (Domingos, 1999; Elkan, 2001; Liu & Zhou, 2006) addresses unequal misclassification costs and class imbalance, which are critical in applications like medical diagnosis, where misclassifying malignant cases as benign can have severe consequences. These algorithms balance class importance by reweighting samples, adjusting decision thresholds, and modifying loss functions (Kukar et al., 1998; Zadrozny et al., 2003; Zhou & Liu, 2010; Khan et al., 2017). In the context of adversarial settings, cost-sensitive learning algorithms have been proposed to mostly empirically improve robustness of classification models such as naive Bayes, linear discriminant, and neural networks (Dalvi et al., 2004; Asif et al., 2015; Dreossi et al., 2018). The only existing method that addresses a cost-sensitive robustness certification problem similar to ours is Zhang & Evans (2019). However, their approach suffers from inherent flexibility issues, making it inapplicable for certifying deep networks under large perturbations, and demonstrates empirically weaker performance compared to ours as demonstrated in Section 6.1.

## 3. Preliminaries

We use lowercase boldfaced letters to denote vectors and uppercase boldfaced letters for matrices. Let $[m]$ be the index set $\{1, 2, ..., m\}$, $|\mathcal{S}|$ be the cardinality of a set $\mathcal{S}$ and $\mathbb{1}(\cdot)$ be the indicator function. For any $\boldsymbol{x} \in \mathbb{R}^d$ and $i \in [d]$, the $i$-th element of $\boldsymbol{x}$ is denoted as $x_i$. Denote by $\mathcal{N}(\boldsymbol{\mu}, \sigma^2\mathbf{I})$ the multivariate spherical Gaussian distribution with mean $\boldsymbol{\mu}$ and covariance matrix $\sigma^2\mathbf{I}$ with $\sigma > 0$. Let $\Phi(\cdot)$ be the cumulative distribution function (CDF) of $\mathcal{N}(0, 1)$ and $\Phi^{-1}(\cdot)$ be its inverse. Let $f_\theta : \mathcal{X} \to [m]$ be a classifier modeled by a DNN with parameter $\theta$.

### 3.1. Adversarial Robustness

Deep neural networks have been shown to be vulnerable to adversarial examples, where an adversary aims to subtly perturb an input $\boldsymbol{x}$ to cause the classifier to make incorrect predictions. Specifically, given a classifier $f_\theta$, a test input $\boldsymbol{x}$ with ground-truth label $y$, and a norm bound $\epsilon > 0$, the adversary seeks to find a perturbation $\boldsymbol{\delta}$ such that:

$$f_\theta(\boldsymbol{x} + \boldsymbol{\delta}) \neq y, \quad \text{where } \|\boldsymbol{\delta}\|_p \leq \epsilon. \tag{1}$$

In this context, $\boldsymbol{\delta}$ represents an adversarial perturbation that is often visually imperceptible to humans but sufficient to deceive the classifier. The magnitude of the perturbation $\|\boldsymbol{\delta}\|_p$ is typically measured in $\ell_2$ or $\ell_\infty$ norm.

To defend against such attacks, a model $f_\theta$ is said to be *provably robust* for an input $\boldsymbol{x}$ if it can be certified that no adversarial perturbation within the allowed norm can change the model's prediction:

$$f_\theta(\boldsymbol{x} + \boldsymbol{\delta}) = f_\theta(\boldsymbol{x}), \; \forall \boldsymbol{\delta} \text{ satisfying } \|\boldsymbol{\delta}\|_p \leq \epsilon. \tag{2}$$

This means that the model's prediction remains unchanged for all possible perturbations $\boldsymbol{\delta}$ within the $\epsilon$-norm ball.

### 3.2. Randomized Smoothing

Randomized smoothing is a probabilistic certification framework proposed in Cohen et al. (2019). In particular, it builds upon the following definition of smoothed classifiers:

**Definition 3.1.** Let $\mathcal{X} \subseteq \mathbb{R}^d$ be the input space and $[m]$ be the label space. For any classifier $f_\theta : \mathcal{X} \to [m]$ and $\sigma > 0$, the *smoothed classifier* with $f_\theta$ and $\sigma$ is defined as:

$$g_\theta(\boldsymbol{x}) = \arg\max_{k \in [m]} \mathbb{P}_{\boldsymbol{\delta} \sim \mathcal{N}(\boldsymbol{0}, \sigma^2\mathbf{I})} \left[ f_\theta(\boldsymbol{x} + \boldsymbol{\delta}) = k \right], \; \forall \boldsymbol{x} \in \mathcal{X}.$$

Let $h_\theta : \mathcal{X} \to [0, 1]^m$ be the corresponding function that maps any $\boldsymbol{x} \in \mathcal{X}$ to the prediction probabilities of $g_\theta(\boldsymbol{x})$:

$$[h_\theta(\boldsymbol{x})]_k = \mathbb{P}_{\boldsymbol{\delta} \sim \mathcal{N}(\boldsymbol{0}, \sigma^2\mathbf{I})} \left[ f_\theta(\boldsymbol{x} + \boldsymbol{\delta}) = k \right], \; \forall k \in [m].$$

The following lemma, proven in Cohen et al. (2019), characterizes the $\ell_2$ perturbation ball with the largest radius for any input $\boldsymbol{x}$ such that the prediction of $g_\theta$ remains the same.

**Lemma 3.2.** Given $(\boldsymbol{x}, y)$, suppose $g_\theta$ classifies $\boldsymbol{x}$ correctly, i.e., $y = \arg\max_{k \in [m]} \mathbb{P}_{\boldsymbol{\delta} \sim \mathcal{N}(\boldsymbol{0}, \sigma^2\mathbf{I})}[f_\theta(\boldsymbol{x} + \boldsymbol{\delta}) = k]$, then $g_\theta$ is provably robust at $\boldsymbol{x}$ in $\ell_2$-norm with the *standard certified radius* $r(\boldsymbol{x}; h_\theta)$ defined by:

$$r(\boldsymbol{x}; h_\theta) = \frac{\sigma}{2} \Big[ \Phi^{-1}\big([h_\theta(\boldsymbol{x})]_y\big) - \Phi^{-1}\big( \max_{k \neq y} [h_\theta(\boldsymbol{x})]_k \big) \Big],$$

where $h_\theta$ is defined in Definition 3.1. If the prediction is incorrect, then $r(\boldsymbol{x}; h_\theta)$ is defined to be 0. When $h_\theta$ is clear from the context, we simply write $r(\boldsymbol{x}) = r(\boldsymbol{x}; h_\theta)$.

## 4. Certifying Cost-Sensitive Robustness

### 4.1. Cost-Sensitive Robustness

We consider image classification tasks under cost-sensitive scenarios, where the goal is to learn a classifier with high cost-sensitive robustness, while maintaining a similar performance level of overall accuracy. Specifically, we define a cost matrix $\mathbf{C} \in \mathbb{R}_{\geq 0}^{m \times m}$ that encodes the potential harm associated with different class-wise adversarial transformations. For any pair of classes $j, k \in [m]$, if $C_{jk} > 0$, it indicates that a misclassification from class $j$ (*seed class*) to class $k$ (*target class*) incurs a non-negligible cost $C_{jk}$. Conversely, $C_{jk} = 0$ suggests that there is no significant incentive for an attacker to induce this specific misclassification.

The goal of cost-sensitive robust learning is to minimize adversarial misclassifications that incur a cost as defined by $\mathbf{C}$. For any *seed class* $y \in [m]$, we let $\Omega_j = \{k \in [m] : C_{jk} > 0\}$ be the set of *cost-sensitive target classes*. If $\Omega_j$ is an empty set, then all the examples from class $j$ are non-sensitive. Otherwise, any class $j$ with $|\Omega_j| \geq 1$ is a sensitive seed class. Given a dataset $\mathcal{S} = \{(\boldsymbol{x}_i, y_i)\}_{i \in [n]}$, we define the set of *cost-sensitive examples* as $\mathcal{S}^s = \{(\boldsymbol{x}, y) \in \mathcal{S} : |\Omega_y| \geq 1\}$, while the remaining examples are regarded as non-sensitive.

### 4.2. Cost-Sensitive Certified Radius

Similar to the standard certified radius, we introduce the definition of the cost-sensitive certified radius. Figure 1 visually illustrates the smoothed classifier and the cost-sensitive certified radius on a cancer classification task.

**Definition 4.1** (Cost-Sensitive Certified Radius)**.** Let $\mathbf{C}$ be an $m \times m$ cost matrix. For any example $(\boldsymbol{x}, y)$, suppose $g_\theta$ classifies $\boldsymbol{x}$ correctly.

1. The *groupwise cost-sensitive certified radius*, denoted as $r_{\text{cs-group}}(\boldsymbol{x}; \Omega_y, h_\theta)$, at $(\boldsymbol{x}, y)$ with the smoothed classifier $g_\theta$ and the cost matrix $\mathbf{C}$ is defined as

$$\frac{\sigma}{2} \Big[ \Phi^{-1}\Big( \max_{k \in [m]} [h_\theta(\boldsymbol{x})]_k \Big) - \Phi^{-1}\Big( \max_{k \in \Omega_y} [h_\theta(\boldsymbol{x})]_k \Big) \Big],$$

where $\Omega_y = \{k \in [m] : C_{yk} > 0\}$ represents the set of sensitive target classes.

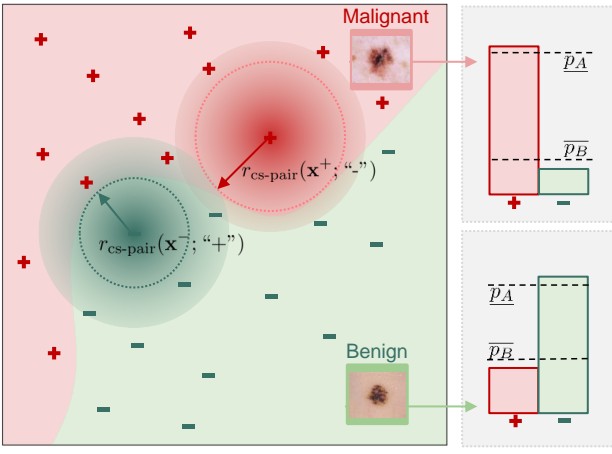

*Figure 1.* A visual demonstration of the evaluation of the smoothed classifier $g_\theta$ and the cost-sensitive certified radius $r_{\text{cs-pair}}$ for the benign/malignant cancer classification task. The right panel illustrates the desired predicted probabilities $h_\theta$ for the benign ($\boldsymbol{x}^-$) and malignant ($\boldsymbol{x}^+$) samples.

2. The *pairwise cost-sensitive certified radius*, denoted as $r_{\text{cs-pair}}(\boldsymbol{x}; j, h_\theta)$, at $(\boldsymbol{x}, y)$ with the smoothed classifier $g_\theta$ and the cost matrix $\mathbf{C}$ is defined as

$$\frac{\sigma}{2}\left[\Phi^{-1}\left(\max_{k\in[m]}\big[h_\theta(\boldsymbol{x})\big]_k\right) - \Phi^{-1}\left(\big[h_\theta(\boldsymbol{x})\big]_j\right)\right],$$

where $j \in \Omega_y$ stands for the misclassification target class.

For incorrect predictions, all certified radii are defined as $0$. For brevity, we write $r_{\text{cs-group}}(\boldsymbol{x}; \Omega_y)$ or $r_{\text{cs-pair}}(\boldsymbol{x}; j)$ if $h_\theta$ is clear from the context.

Using similar proof techniques as in Lemma 3.2, we can prove that no adversarial samples for $\boldsymbol{x}$ exist within the $\ell_2$ radius $r_{\text{cs-group}}(\boldsymbol{x}; \Omega_y)$ that introduce misclassification costs. Likewise, there are no targeted adversarial samples that misclassify $\boldsymbol{x}$ to class $j$ within the $\ell_2$ radius $r_{\text{cs-pair}}(\boldsymbol{x}; j)$. Note that the standard certified radius $r(\boldsymbol{x})$ defined in Cohen et al. (2019) also serves as a valid (though conservative) certificate under cost-sensitive settings. The following theorem characterizes the relationship between the cost-sensitive and standard certified radii.

**Theorem 4.2.** Under the same setting as in Definition 4.1, suppose that $\arg\max_{k\in[m]}[h_\theta(\boldsymbol{x})]_k \notin \Omega_y$. Then, we have $r_{\text{cs-pair}}(\boldsymbol{x}; j) \geq r_{\text{cs-group}}(\boldsymbol{x}; \Omega_y) \geq r(\boldsymbol{x})$, where the first equality holds when $\Omega_y = \{j\}$ and the second holds when $\Omega_y = \{k \in [m] : k \neq y\}$.

The key distinction between $r_{\text{cs-group}}(\boldsymbol{x}; \Omega_y)$ and the standard certified radius $r(\boldsymbol{x})$ arises when $|\Omega_y| < m - 1$ (see Appendix B.1 for detailed proofs of Theorem 4.2). Notably, the benefit of using the cost-sensitive certified radius is more significant for $(\boldsymbol{x}, y)$ with smaller $|\Omega_y|$, as $\Phi^{-1}$ is monotonically increasing, and the gap between $[h_\theta(\boldsymbol{x})]_y$

---

**Algorithm 1** Certification for Cost-Sensitive Robustness

1: **function** CERTIFY_GROUP($f_\theta, \sigma, \boldsymbol{x}, n_0, n, \alpha, \Omega_y$) :
2:   //Determine the top-1 class
3:   counts0 $\leftarrow$ SampleUnderNoise($f_\theta, \boldsymbol{x}, n_0, \sigma$)
4:   $\hat{c}_A \leftarrow$ top index in counts0
5:   //Obtain the counts
6:   counts $\leftarrow$ SampleUnderNoise($f_\theta, \boldsymbol{x}, n, \sigma$)
7:   //Compute the standard radius
8:   $\hat{r}_{\text{std}} = \Phi^{-1}(\text{LCB}(\text{counts}[\hat{c}_A], n, 1-\alpha)) \cdot \sigma$
9:   //Compute the group-wise radius
10:   $\underline{p_A} = \text{LCB}(\text{counts}[\hat{c}_A], n, 1-\frac{\alpha}{2})$
11:   $\overline{p_B} = \max\{\text{UCB}(\text{counts}[k], n, 1-\frac{\alpha}{2|\Omega_y|}) : k \in \Omega_y\}$
12:   $\hat{r}_{\text{cs-group}} = (\Phi^{-1}(\underline{p_A}) - \Phi^{-1}(\overline{p_B})) \cdot \sigma/2$
13:   //Return the final result
14:   **if** $\max(\hat{r}_{\text{std}}, \hat{r}_{\text{cs-group}}) > 0$ **then**
15:     **return** $\hat{c}_A, \max(\hat{r}_{\text{std}}, \hat{r}_{\text{cs-group}})$
16:   **else return** ABSTAIN

---

and $\max_{k\in\Omega_y}[h_\theta(\boldsymbol{x})]_k$ tends to be larger with a smaller $\Omega_y$ (see Figures 3(b) and 3(c) for empirical evidence). Although the pairwise certified radius $r_{\text{cs-pair}}(\boldsymbol{x}; j)$ provides weaker guarantees (addressing only specific targeted misclassifications), it enables defining an overall robust cost (Definition 4.3), offering fine-grained control to mitigate critical misclassifications defined by the cost matrix (Section 5).

**Definition 4.3** (Cost-Sensitive Robust Cost). Given the cost-sensitive certified radii in Definition 4.1, the overall objective of cost-sensitive robust learning can be formulated as minimizing the total *certified robust cost*:

$$\text{Rob}_{\text{cost}}(g_\theta) = \frac{1}{|\mathcal{S}^s|} \sum_{(\mathbf{x},y)\in\mathcal{S}^s} \sum_{j\in\Omega_y} C_{yj} \cdot \mathbb{1}\big\{r_{\text{cs-pair}}(\mathbf{x}; j) \leq \epsilon\big\},$$

where $\epsilon$ is the norm bound for the perturbations. This metric captures the total cost of potential misclassifications for sensitive test samples, with each misclassification cost defined by the cost matrix $\mathbf{C}$.

### 4.3. Proposed Certification Algorithm

Since exact computation of the cost-sensitive certified radius requires infinitely many Gaussian samples, we adopt Monte Carlo methods (Cohen et al., 2019) to obtain its empirical estimates. The main challenge for our adaptation lies in how to ensure statistically rigorous and tighter bounds on the newly proposed cost-sensitive certified radius, as detailed in Algorithm 1 (see Algorithm 2 for CERTIFY_PAIR).

Following standard practice, we first compute the $(1-\alpha)$ lower confidence bound for the cost-sensitive certified radius by calculating $\hat{r}_{\text{cs}}$ such that $\mathbb{P}[\hat{r}_{\text{cs}} \leq r_{\text{cs}}] \geq 1-\alpha$. This ensures $\hat{r}_{\text{cs}}$ is a valid lower bound on the true radius $r_{\text{cs}}$ with high probability, providing rigorous certification even

if conservative. We now introduce our approach to improve the tightness of this estimation. Recall that the certified radius has the following general form:

$$\frac{\sigma}{2}\big[\Phi^{-1}(p_A) - \Phi^{-1}(p_B)\big].$$

To lower-bound the radius, we estimate $\Phi^{-1}(p_A)$ and $\Phi^{-1}(p_B)$ by computing $\underline{p_A}$ (a lower bound for $p_A$) and $\overline{p_B}$ (an upper bound for $p_B$):

- *Lower-bounding $p_A$*: Compute $\underline{p_A}$ as the $(1-\alpha)$ lower confidence bound of $p_A$, and set $\overline{p_B} = 1-\underline{p_A}$. This yields the lower bound radius $\sigma\Phi^{-1}(\underline{p_A})$, as $p_B \leq 1-\underline{p_A}$. This method, used in Cohen et al. (2019), gives the empirical radius $\hat{r}_{\text{std}}$, which, as shown in Theorem 4.2, is always a valid $(1-\alpha)$ lower bound for the cost-sensitive radius.

- *Computing both bounds for $p_A$ and $p_B$*: The radius can also be computed using a $(1-\alpha/2)$ lower confidence bound of $p_A$ and a $(1-\alpha/2)$ upper confidence bound of $p_B$. For $\hat{r}_{\text{cs-group}}$, the $(1-\alpha/2)$ upper confidence bound of $p_B$ is further approximated by taking the maximum of the $(1-\alpha/(2|\Omega_y|))$ upper confidence bounds for each class $k \in \Omega_y$. For $\hat{r}_{\text{cs-pair}}$, $\overline{p_B}$ is directly computed as the $(1-\alpha/2)$ upper confidence bound of the target class $j$.

The following theorem, proven in Appendix B.2 by union bound, shows the validity of the estimate $\hat{r}_{\text{cs}}$ as a $(1 - \alpha)$ confidence bound on the certified cost-sensitive radius.

**Theorem 4.4.** For any example $(\boldsymbol{x}, y)$, smoothed classifier $g_\theta$ and cost matrix $\mathbf{C}$, $\hat{r}_{\text{cs-group}}$ and $\hat{r}_{\text{cs-pair}}$ are certified cost-sensitive robust radii with at least $(1-\alpha)$ confidence over the randomness of Gaussian sampling.

*Remark* 4.5. Unlike standard certified radius derivations, our algorithm introduces a novel method for computing the upper confidence bound $\overline{p_B}$ over cost-sensitive target classes $\Omega_y$. Standard approaches (Cohen et al., 2019) set $\overline{p_B} = 1-\underline{p_A}$, assuming concentration of non-ground-truth class probabilities in a single runner-up class. However, this is sub-optimal for certifying cost-sensitive robustness, as the relevant top class in $\Omega_y$ (for groupwise radius) or target class $j$ (for pairwise radius) often differs from the runner-up, yielding a looser robustness certificate. Our method refines $\overline{p_B}$ using union bounds and tailored confidence levels, yielding tighter certificates (e.g., Line 11, Algorithm 1).

*Remark* 4.6. Our certification algorithm returns the maximum between the two possible radii $\hat{r}_{\text{std}}$ and $\hat{r}_{\text{cs}}$, ensuring the tighter bound is returned. While the theoretical results suggest that $r_{\text{cs-pair}} \geq r_{\text{cs-group}} \geq r$, empirical cases may show

$\hat{r}_{\text{std}} > \hat{r}_{\text{cs}}$, especially when $\Omega_y$ includes the runner-up class. Overall, the gap between $\hat{r}_{\text{std}}$ and $\hat{r}_{\text{cs}}$ depends on input-specific factors, such as the probabilities of the top and runner-up classes, as well as hyperparameters like $n$ and $\alpha$ (see visualizations and discussions in Appendix B.3).

## 5. Training for Cost-Sensitive Robustness

While the notion of cost-sensitive certified robustness (Section 4.2) and our certification algorithms (Section 4.3) can already be applied independently to any classifier in a black-box manner, this section focuses on training methods designed to enhance robustness guarantees in these contexts.

**Margin-CS: Cost-sensitive Margin Loss.** We consider the natural idea of minimizing the total cost-sensitive robust cost (Definition 4.3) directly. However, this is challenging because the certified radius is not differentiable (nor sub-differentiable) due to the presence of $\arg\max$ operations and counting procedures, making it unsuitable for direct incorporation into the objective function. To address this, we employ a soft-smoothed classifier approximation for the certified radius, following approaches in Salman et al. (2019); Zhai et al. (2020); Jeong et al. (2021). This replaces the counting probability $h_\theta$ with the soft-smoothed prediction $G_\theta(\boldsymbol{x}) = \mathbb{E}_{\boldsymbol{\delta} \sim \mathcal{N}(0, \sigma^2 I)}[F_\theta(\boldsymbol{x} + \boldsymbol{\delta})]$ when computing the radius, where $F_\theta : \mathbb{R}^d \to \Delta^{m-1}$ is the soft-base classifier that returns the predicted probability scores (see Appendix A for notation summary). We denote this approximation as $r_{\text{cs-pair}}(\boldsymbol{x}; j, G_\theta)$ and $r_{\text{cs-group}}(\boldsymbol{x}; j, G_\theta)$, emphasizing the use of the soft-smoothed classifier $G_\theta$. In addition, maximizing the certified radius can be interpreted as increasing the margin between the smoothed classifier's confidence in the ground-truth class and the most confusing target class.

To achieve numerical stability in this process, we apply hinge loss, which mitigates the instability caused by the large derivative of the inverse cumulative density function $\Phi^{-1}(\boldsymbol{x})$ near the domain boundary, as also highlighted in Zhai et al. (2020). More formally, given thresholds $l \leq u$, the general margin loss for any $v \in \mathbb{R}$ is defined as:

$$\mathcal{L}_{\text{M}}(v; l, u) = \max\{u - v, 0\} \cdot \mathbb{1}\Big[l \leq v \leq u\Big], \quad (3)$$

where the indicator function selects data points whose certified radius falls within the range of $[l, u]$. The overall training objective of our method is defined as:

$$\min_{\theta \in \Theta} \Bigg\{ \mathbb{E}_{(\boldsymbol{x},y) \sim \mathcal{D}} \mathbb{E}_{\boldsymbol{\delta} \sim \mathcal{N}(0, \sigma^2 \mathbf{I})} \mathcal{L}_{\text{CE}}(f_\theta(\boldsymbol{x} + \boldsymbol{\delta}), y)$$

$$+ \lambda_1 \Big( \mathbb{E}_{(\boldsymbol{x},y) \sim \mathcal{D}_n} \mathcal{L}_{\text{M}}(r_{\text{cs-group}}(\boldsymbol{x}; \Omega_y, G_\theta); 0, \gamma_1) \quad (4)$$

$$+ \lambda_2 \mathbb{E}_{(\boldsymbol{x},y) \sim \mathcal{D}_s} \sum_{j \in \Omega_y} C_{yj} \mathcal{L}_{\text{M}}(r_{\text{cs-pair}}(\boldsymbol{x}; j, G_\theta); 0, \gamma_2) \Big) \Bigg\}.$$

Here, $\lambda_1, \lambda_2, \gamma_1, \gamma_2 > 0$ are hyperparameters, $\mathcal{D}$ denotes the overall data distribution, $\mathcal{D}_s$ is the distribution of sensitive examples that incur costs when misclassified, and $\mathcal{D}_n$ is the distribution of normal examples. Specifically, the first term focuses on model performance, while the last two terms focus on robustness, with the final term incorporating the approximated total robust cost. The interval defined by $\gamma_1$ and $\gamma_2$ in the margin loss $\mathcal{L}_M$ determines which data subpopulation is prioritized during optimization. Larger values of $\gamma_1$ and $\gamma_2$ result in broader data coverage (see Appendix D.3 for detailed ablation studies).

Note that by applying different threshold restrictions to the certified radius for sensitive and non-sensitive samples, the model can focus on optimizing specific data subpopulations rather than treating all data points equally. This targeted approach is particularly beneficial for cost-sensitive learning, with our experiments showing that this fine-grained optimization strategy improves certified cost-sensitive robustness without compromising overall standard accuracy.

# 6. Experiments

**Datasets & Configurations.** We evaluate our method on the standard benchmark datasets: CIFAR-10 (Krizhevsky et al., 2009), Imagenette[1], and the full ImageNet dataset (Deng et al., 2009). In addition, we assess its performance on the real-world medical dataset HAM10k (Tschandl et al., 2018) to demonstrate its effectiveness in practical scenarios, where cost-sensitive misclassifications can have severe consequences. For CIFAR-10 and HAM10k, we use the ResNet architecture following Cohen et al. (2019) as the target classification model. Specifically, we use ResNet-56, since it attains a comparable performance to ResNet-110 but with reduced computation costs. For ImageNet, we use ResNet-18, following Pethick et al. (2023). Consistent with common evaluation practices (Cohen et al., 2019), we focus on the setting of $\epsilon = 0.5$ and $\sigma = 0.5$ in our experiments, while we observe similar trends under other settings (see Appendix D for all the additional experimental results).

**Methods.** We compare our *Margin-CS* with existing baseline randomized smoothing-based training methods, including the Gaussian augmentation-based method (Cohen et al., 2019) (denoted as *Gaussian*), *SmoothAdv* (Salman et al., 2019), *SmoothMix* (Jeong et al., 2021), and *MACER* (Zhai et al., 2020), which are originally proposed to optimize for overall robustness. For reference, we also adapt these methods for cost-sensitive scenarios (when applicable), introducing *Gaussian-CS*, *SmoothAdv-CS*, and *SmoothMix-CS* as adaptive baselines (see Appendix D.1 for detailed descriptions). These adaptations use reweighting, a common technique in cost-sensitive learning, and are optimized for

cost-sensitive performance by tuning the weight parameters associated with sensitive examples.

**Evaluation Metrics.** We evaluate the performance of different methods using the following metrics: (i) *Certified robust cost* (Definition 4.3), (ii) *Certified cost-sensitive robustness*: $\mathrm{Rob}_{\mathrm{cs}}(g_\theta) = \frac{1}{|\mathcal{S}^s|} \sum_{(\boldsymbol{x},y) \in \mathcal{S}^s} \mathbb{1}\{r_{\mathrm{cs\text{-}group}}(\boldsymbol{x}; \Omega_y) > \epsilon\}$, which measures the fraction of (cost-)sensitive test examples with a certified radius larger than $\epsilon$, indicating robustness under $\ell_2$ perturbations, where $\mathcal{S}^s$ is the set of sensitive test examples, and (iii) *Certified overall accuracy*: $\mathrm{Acc}(g_\theta) = \frac{1}{|\mathcal{S}|} \sum_{(x,y) \in \mathcal{S}} \mathbb{1}\{r(\boldsymbol{x}) > 0\}$, which computes the ratio of correctly classified samples by $g_\theta$ over the entire testing dataset. Using the above metrics, we can systematically compare the cost-sensitive robustness and standard performance of various robust training methods.

## 6.1. Main Results

**Equal Costs for Selected Misclassifications.** We start by evaluating a straightforward scenario on standard benchmarks where equal costs are assigned to specific misclassification cases. This demonstrates that our approach can effectively emphasize these critical misclassifications. Specifically, we examine the following cases: *"S-Seed"*: A single randomly selected sensitive seed class, where misclassification to any other class is assigned an equal cost. *"M-Seed"*: Multiple sensitive seed classes, where misclassification to any other class is assigned an equal cost. *"S-Pair"*: A single misclassification from a sensitive seed class to one specific target class. *"M-Pair"*: A single sensitive seed class with misclassifications to multiple target classes, each with an equal cost. In our experiments, we report performance on classes that typically show higher prediction error by a vanilla classifier (i.e., those more vulnerable to attacks).

Tables 1 and 2 show that Margin-CS significantly surpasses all standard baselines, improving by approximately 20% over the best standard robust training method while maintaining overall certified accuracy compared to non-cost-sensitive baselines. Beyond the quantitative results, a visual demonstration of the distributions of the certified radii in Figure 3(a) (CIFAR-10 *"S-Seed"* case) shows that our Margin-CS generally increases the certified radii for most samples. These results indicate that our fine-grained thresholding techniques for optimizing the certified radius offer a better trade-off between accuracy and cost-sensitive robustness than both non-adaptive baselines and standard adaptations.

**Comparisons with Existing Methods.** We compare our method with Zhang & Evans (2019)'s, which, to the best of our knowledge, is the only existing work that provides a certification and training approach for cost-sensitive robustness. While Zhang & Evans (2019) does not provide a certified radius—potentially favoring our method due to its

[1] https://github.com/fastai/Imagenette.

*Table 1.* Certification results on CIFAR-10 for selected misclassifications (*S-Seed*, *M-Seed*, *S-Pair*, and *M-Pair*) with equal costs. **Acc** is *certified overall accuracy* (%), $\mathbf{Rob_{cs}}$ refers to *certified cost-sensitive robustness* (%), and $\mathbf{Rob_{cost}}$ denotes *certified robustness cost*.

| Method | S-Seed | | | M-Seed | | | S-Pair | | | M-Pair | | |
|---|---|---|---|---|---|---|---|---|---|---|---|---|
| | Acc ↑ | $Rob_{cs}$ ↑ | $Rob_{cost}$ ↓ | Acc ↑ | $Rob_{cs}$ ↑ | $Rob_{cost}$ ↓ | Acc ↑ | $Rob_{cs}$ ↑ | $Rob_{cost}$ ↓ | Acc ↑ | $Rob_{cs}$ ↑ | $Rob_{cost}$ ↓ |
| Gaussian | 65.4 | 22.3 | 4.99 | 65.9 | 29.3 | 4.67 | 65.4 | 50.4 | 0.18 | 65.4 | 33.6 | 0.92 |
| SmoothMix | 65.7 | 17.2 | 6.29 | 65.7 | 26.4 | 4.61 | 65.7 | 58.0 | 0.39 | 65.7 | 45.2 | 1.46 |
| SmoothAdv | 66.9 | 27.1 | 4.94 | 66.9 | 30.2 | 4.42 | 66.9 | 57.1 | 0.38 | 66.9 | 38.7 | 1.23 |
| MACER | 65.9 | 27.3 | 5.27 | 65.9 | 29.1 | 4.90 | 65.8 | 54.3 | 0.23 | 65.8 | 38.5 | 0.99 |
| Gaussian-CS | 64.2 | 50.6 | 3.35 | 66.2 | 34.8 | 4.16 | 64.2 | 72.3 | 0.08 | 64.2 | 64.3 | 0.48 |
| SmoothMix-CS | 63.2 | 26.3 | 5.25 | 65.0 | 29.5 | 4.19 | 63.2 | 67.5 | 0.20 | 63.2 | 46.2 | 1.17 |
| SmoothAdv-CS | 66.1 | 53.5 | 3.12 | 67.2 | 43.3 | 3.14 | 66.1 | 80.4 | 0.31 | 66.1 | 75.3 | 0.73 |
| Margin-CS | **67.5** | **54.8** | **3.04** | **67.3** | **46.8** | **3.07** | **67.3** | **92.4** | **0.05** | **67.5** | **80.4** | **0.35** |

*Table 2.* Certification results on Imagenette for selected misclassifications (*S-Seed*, *M-Seed*, *S-Pair*, and *M-Pair*) with equal costs. **Acc** is *certified overall accuracy* (%), $\mathbf{Rob_{cs}}$ refers to *certified cost-sensitive robustness* (%), and $\mathbf{Rob_{cost}}$ denotes *certified robustness cost*.

| Method | S-Seed | | | M-Seed | | | S-Pair | | | M-Pair | | |
|---|---|---|---|---|---|---|---|---|---|---|---|---|
| | Acc ↑ | $Rob_{cs}$ ↑ | $Rob_{cost}$ ↓ | Acc ↑ | $Rob_{cs}$ ↑ | $Rob_{cost}$ ↓ | Acc ↑ | $Rob_{cs}$ ↑ | $Rob_{cost}$ ↓ | Acc ↑ | $Rob_{cs}$ ↑ | $Rob_{cost}$ ↓ |
| Gaussian | 80.3 | 64.6 | 3.67 | 80.4 | 58.9 | 3.86 | 80.3 | 88.5 | 0.236 | 80.3 | 75.6 | 1.09 |
| SmoothMix | 80.2 | 64.3 | 3.91 | 80.2 | 55.5 | 4.86 | 80.2 | 89.4 | 0.198 | 80.2 | 79.2 | 1.76 |
| SmoothAdv | **80.6** | 59.5 | 2.93 | 80.6 | 59.2 | 3.98 | 80.6 | 90.2 | 0.196 | **80.6** | 74.9 | 1.72 |
| MACER | 78.2 | 63.8 | 2.46 | 78.2 | 57.8 | 2.72 | 78.2 | 89.9 | 0.169 | 78.2 | 78.0 | 0.34 |
| Gaussian-CS | 74.6 | 73.3 | 1.67 | 74.0 | 61.7 | 2.52 | 75.4 | 91.1 | 0.167 | 75.4 | 83.0 | 0.26 |
| SmoothMix-CS | 77.6 | 66.6 | 3.82 | 76.1 | 68.9 | 4.30 | 77.6 | 90.2 | 0.189 | 77.3 | 82.6 | 1.71 |
| SmoothAdv-CS | 76.1 | 68.9 | 2.24 | 75.7 | 62.6 | 2.81 | 78.6 | 90.7 | 0.177 | 77.6 | 80.4 | 1.70 |
| Margin-CS | 79.6 | **81.1** | **1.35** | **82.1** | **72.0** | **2.46** | **82.7** | **94.7** | **0.167** | 79.6 | **86.3** | **0.21** |

*Table 3.* Comparisons of our method with Zhang & Evans (2019) for $\ell_2$-norm on CIFAR-10 across different settings.

| Method | S-Pair | | M-Pair (1,1,10) | | M-Pair (1,1,2) | |
|---|---|---|---|---|---|---|
| | Acc↑ | $Rob_{cs}$↑ | Acc↑ | $Rob_{cost}$↓ | Acc↑ | $Rob_{cost}$↓ |
| Gaussian | 79.3 | 67.4 | 79.3 | 5.04 | 79.3 | 2.02 |
| SmoothMix | 73.0 | 67.0 | 73.0 | 3.98 | 73.0 | 2.32 |
| SmoothAdv | 77.9 | 74.2 | 77.9 | 4.45 | 77.9 | 2.12 |
| MACER | 78.4 | 77.4 | 78.4 | 4.52 | 78.4 | 1.35 |
| Gaussian-CS | 77.8 | 81.1 | 77.8 | 4.21 | 77.8 | 1.27 |
| SmoothMix-CS | 72.2 | 75.7 | 72.2 | 2.65 | 72.2 | 1.97 |
| SmoothAdv-CS | 78.7 | 83.8 | 78.7 | 3.21 | 78.7 | 1.43 |
| Zhang & Evans (2019) | 61.2 | 92.4 | 78.0 | 2.78 | 73.9 | 1.70 |
| Margin-CS | **80.9** | **93.5** | 79.5 | **2.39** | 79.5 | **0.93** |

*Table 4.* Performance metrics (Accuracy, Precision, Recall in %) and certification results ($\mathbf{Rob_{cs}}$ in % and $\mathbf{Rob_{cost}}$) on HAM10k.

| Method | HAM10k | | | | |
|---|---|---|---|---|---|
| | Acc ↑ | $Rob_{cs}$ ↑ | $Rob_{cost}$ ↓ | Precision ↑ | Recall ↑ |
| Gaussian | 82.9 | 11.8 | 1.56 | 51.0 | 15.0 |
| SmoothAdv | 82.6 | 11.4 | 1.68 | 36.4 | 17.8 |
| SmoothMix | 83.1 | 0.79 | 1.66 | 40.0 | 0.80 |
| MACER | 82.7 | 21.1 | 1.41 | 50.0 | 25.0 |
| Gaussian-CS | 80.5 | 19.7 | 1.47 | 38.0 | 20.0 |
| SmoothAdv-CS | 81.8 | 21.7 | 1.64 | 39.6 | 23.5 |
| SmoothMix-CS | 81.9 | 2.1 | 1.70 | 25.0 | 2.1 |
| Margin-CS | **83.2** | **34.4** | **1.17** | **52.0** | **41.3** |

flexibility—all our evaluation metrics remain valid. Since their default setting aligns close to $\epsilon = 0.25$, we perform all comparisons with both training and testing noise set to $\sigma = 0.25$. The comparison results in Table 3 focus on seed class "3" under both equal cost (the *"S-Pair"* setting) and varying unequal cost conditions (the *"M-Pair (1,1,10)"* and *"M-Pair (1,1,2)"* settings). In the unequal cost scenarios, the misclassification costs for class "3" to classes "2", "4", "5" are set to 1, 1, and 10, and to classes "2", "4", "6" are set to 1, 1, and 2, respectively.

As shown in Table 3, our Margin-CS achieves significantly higher certified robustness while maintaining comparable overall standard accuracy when compared with Zhang & Evans (2019), highlighting the superior practicality of our approach. It is important to note that, as a convex relaxation-based method, Zhang & Evans (2019) is not feasible for relatively large real-world datasets (e.g., Imagenette and HAM10k in our work), limiting its applicability.

## 6.2. Applicability to Real-World Medical Data

We consider a practically significant scenario of cost-sensitive robust learning using real-world medical data. Specifically, we evaluate the performance of various robust training methods on the HAM10k (Tschandl et al., 2018) dataset, focusing on the task of classifying images of pigmented lesions as either benign or malignant. This scenario closely mirrors real-world applications, where misclassifying a malignant tumor as benign can lead to severe consequences and higher associated costs. To account for this, we set the misclassification costs at a ratio of 10:1 for malignant-to-benign versus benign-to-malignant errors, emphasizing the significantly higher penalty for failing to diagnose a malignant condition.

The results, shown in Table 4, include both performance metrics (Accuracy, Precision, Recall) for the smooth classifier, as well as certification metrics ($\mathbf{Rob_{cost}}$ and $\mathbf{Rob_{cs}}$). Notably, our Margin-CS method, which directly optimizes

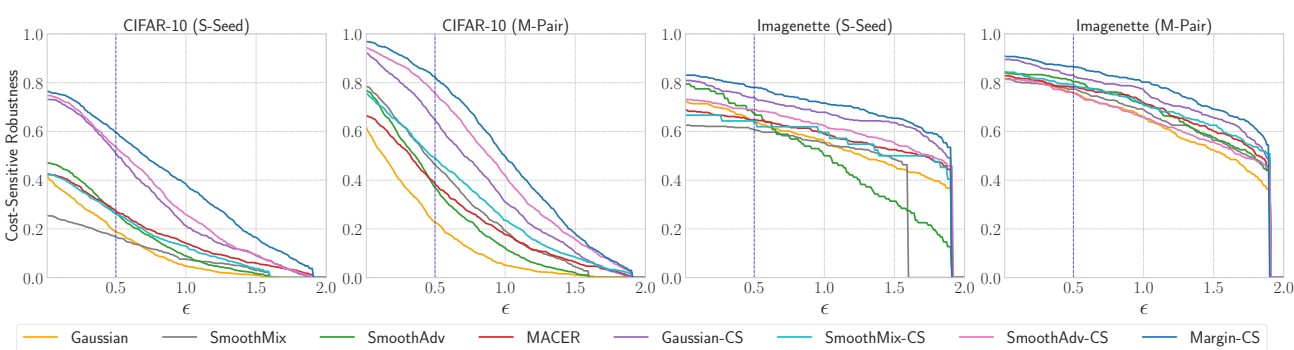

*Figure 2.* Certified accuracy curves for varying $\epsilon$ on different datasets under different settings.

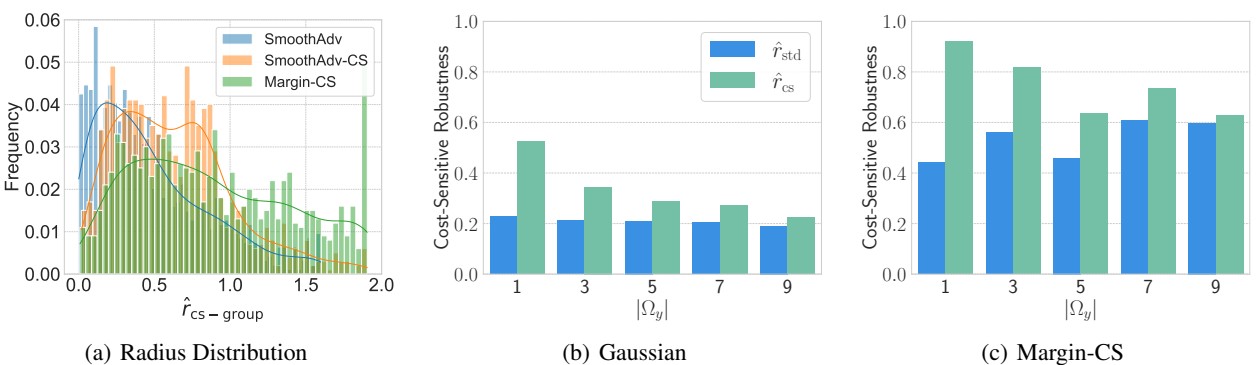

(a) Radius Distribution          (b) Gaussian          (c) Margin-CS

*Figure 3.* Figure 3(a) shows Margin-CS improves the distribution of cost-sensitive certified radius compared with the overall strongest baselines, SmoothAdv and SmoothAdv-CS. Figures 3(b) and 3(c) compare the cost-sensitive robustness computed using $\hat{r}_{\text{std}}$ and $\hat{r}_{\text{cs-group}}$.

the certified radius, shows significant improvements in certified robustness and robust cost. We also notice that while all methods demonstrate comparable prediction accuracy, they exhibit substantial differences in precision and recall. The intrinsic imbalance in the data distribution (i.e., a higher prevalence of benign samples) leads many training methods to bias predictions toward benign outcomes. Notably, our Margin-CS achieves superior recall (which is of great practical importance as it captures potentially malignant cases that require urgent treatment) and precision compared to other cost-sensitive robust training methods as well as non-cost-sensitive baselines. For reference, a standard base classifier yields 3.4% precision and 39.3% recall, while our Margin-CS method substantially improves upon these metrics to 52% precision and 41.3% recall.

### 6.3. Ablation Studies with Varying $\epsilon$

To provide a more comprehensive understanding of each method's behavior across varying levels of $\epsilon$ (which correspond to different practical robustness requirements), we compare the cost-sensitive robustness $\mathbf{Rob}_{\text{cs}}$ under varying scale of $\ell_2$ perturbations for all methods in Figure 2. The performance at $\epsilon = 0$ measures the *certified accuracy* for cost-sensitive examples, and the results at $\epsilon = 0.5$ corre-

spond to the default setting we adopted for comparisons. It is evident that our Margin-CS generally outperform all the baseline methods and their adaptations in certified cost-sensitive robustness across different $\epsilon$. The most significant improvements provided by our cost-sensitive adaptation occur for $\epsilon \leq 1$, with notable gains up to $\epsilon \approx 1.5$. Beyond this, larger $\epsilon$ values likely exceed the limits of certification methods, causing failures across all approaches.

## 7. Conclusion

In this work, we developed a comprehensive framework based on randomized smoothing to certify and train for cost-sensitive robustness. We investigate various representative training methods, ranging from systematic adaptations of existing approaches to our novel targeted cost-sensitive certified radius maximization technique. Notably, our fine-grained thresholding techniques, which optimize the certified radius across carefully calibrated data subgroups, significantly improve the model utility-robustness trade-off. Extensive experiments on benchmark and medical datasets demonstrate the generality and effectiveness of our framework compared to existing methods.

## Availability

To ensure reproducibility and accessibility, our method and the implementations of our experiments are available as open source code at: https://github.com/AppleXY/Cost-Sensitive-RS.

## Impact Statement

Our work aims to enhance the robustness of machine learning systems under cost-sensitive scenarios, which is crucial for safety-critical applications such as healthcare and autonomous driving. We are not aware of any ethical issues that might be induced by our work, since our main focus is to build better defenses. As for potential societal consequences, we believe our work will be particularly helpful for practitioners to incorporate domain knowledge to build better robust machine learning systems that are prioritized to defend against the most consequential attacks.

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

# Appendix

## A. Notations

We present below in Table 5 a summary of the notations used throughout this paper. $\mathcal{D}$ represents the underlying data distribution, $\mathcal{D}_s$ is the distribution of all sensitive examples that incur costs if misclassified, and $\mathcal{D}_n$ is the distribution of the remaining normal examples. The base classifier is denoted by $f_\theta : \mathbb{R}^d \to [m]$, while $F_\theta : \mathbb{R}^d \to \Delta^{m-1}$ refers to the soft-base classifier that outputs predicted probability scores, where $\Delta^{m-1}$ is the probability simplex in $\mathbb{R}^m$. Specifically, $f_\theta(\boldsymbol{x}) = \arg\max_{c \in [m]} [F_\theta(\boldsymbol{x})]_c$.

In addition, $g_\theta : \mathbb{R}^d \to [m]$ denotes the smoothed classifier, and $G_\theta : \mathbb{R}^d \to \Delta^{m-1}$ represents the soft-smoothed classifier with $G_\theta(\boldsymbol{x}) = \mathbb{E}_{\boldsymbol{\delta} \sim \mathcal{N}(0,\sigma^2 \mathbf{I})} [F_\theta(\boldsymbol{x} + \boldsymbol{\delta})]$, which serves as an approximation of $h_\theta$ introduced in Definition 3.1. In essence, $h_\theta$ captures the frequency of hard label predictions given by the base classifier $f$ on noisy inputs $\boldsymbol{x} + \boldsymbol{\delta}$, while $G_\theta$ represents the expected soft label predictions produced by the soft-base classifier $F$ on noisy inputs.

*Table 5.* Summary of Notations.

| Notation | Description |
|---|---|
| $\mathcal{D}$ | Overall data distribution |
| $\mathcal{D}_s$ | Distribution of all sensitive examples |
| $\mathcal{D}_n$ | Distribution of the remaining normal examples |
| $f_\theta : \mathbb{R}^d \to [m]$ | The base classifier |
| $F_\theta : \mathbb{R}^d \to \Delta^{m-1}$ | The soft-base classifier |
| $g_\theta : \mathbb{R}^d \to [m]$ | The smoothed classifier |
| $G_\theta : \mathbb{R}^d \to \Delta^{m-1}$ | The soft-smoothed classifier |
| $h_\theta : \mathbb{R}^d \to \Delta^{m-1}$ | The prediction probabilities of $g_\theta$ |
| $\boldsymbol{x} \in \mathbb{R}^d$ | The query sample |
| $\tilde{\boldsymbol{x}} \in \mathbb{R}^d$ | The $\ell_2$-bounded adversarial sample of $\boldsymbol{x}$ on $g_\theta$ |
| $\tilde{\boldsymbol{x}}' \in \mathbb{R}^d$ | The unrestricted adversarial sample of $\boldsymbol{x}$ on $g_\theta$ |

## B. Certification Algorithm

Algorithm 2 provides the pseudocode for computing the pair-wise certified radius, as referenced in Section 4 of the main paper.

### B.1. Proof of Theorem 4.2

*Proof of Theorem 4.2.* Note that $\boldsymbol{x}$ is assumed to be correctly classified by $g_\theta$ in the definition of certified radius (otherwise the radius is defined to be 0), suggesting that the ground-truth $y$ is also the top-1 class. Recall that standard certified radius is defined as:

$$r(\boldsymbol{x}) = \frac{\sigma}{2} \Big[ \Phi^{-1}\big([h_\theta(\boldsymbol{x})]_y\big) - \Phi^{-1}\big(\max_{k \neq y} [h_\theta(\boldsymbol{x})]_k\big)\Big]. \quad (5)$$

Recall that our *cost-sensitive certified radius* is defined as:

$$r_{\text{cs-group}}(\boldsymbol{x}; \Omega_y) = \frac{\sigma}{2}\Big[\Phi^{-1}\big([h_\theta(\boldsymbol{x})]_y\big) - \Phi^{-1}\big(\max_{k \in \Omega_y} [h_\theta(\boldsymbol{x})]_k\big)\Big],$$

---

**Algorithm 2** Certification for Cost-Sensitive Robustness

1: **function** CERTIFY_PAIR($f_\theta, \sigma, \boldsymbol{x}, n_0, n, \alpha, j$) :
2:   //Determine the top-1 class
3:   counts0 $\leftarrow$ SAMPLEUNDERNOISE($f_\theta, \boldsymbol{x}, n_0, \sigma$)
4:   $\hat{c}_A \leftarrow$ top index in counts0
5:   //Obtain the counts
6:   counts $\leftarrow$ SAMPLEUNDERNOISE($f_\theta, \boldsymbol{x}, n, \sigma$)
7:   //Compute the standard radius
8:   $\hat{r}_{\text{std}} = \Phi^{-1}(\text{LCB}(\text{counts}[\hat{c}_A], n, 1 - \alpha)) \cdot \sigma$
9:   //Compute the pair-wise radius
10:   $\underline{p_A} = \text{LCB}(\text{counts}[\hat{c}_A], n, 1 - \frac{\alpha}{2})$
11:   $\overline{p_B} = \text{UCB}(\text{counts}[j], n, 1 - \frac{\alpha}{2})$
12:   $\hat{r}_{\text{cs-pair}} = (\Phi^{-1}(\underline{p_A}) - \Phi^{-1}(\overline{p_B})) \cdot \sigma/2$
13:   //Return the final result
14:   **if** $\max(\hat{r}_{\text{std}}, \hat{r}_{\text{cs-pair}}) > 0$ **then**
15:     **return** $\hat{c}_A, \max(\hat{r}_{\text{std}}, \hat{r}_{\text{cs-pair}})$
16:   **else return** ABSTAIN

---

$$r_{\text{cs-pair}}(\boldsymbol{x}; j) = \frac{\sigma}{2}\Big[\Phi^{-1}\big([h_\theta(\boldsymbol{x})]_y\big) - \Phi^{-1}\big([h_\theta(\boldsymbol{x})]_j\big)\Big].$$

Note that the first term is identical across all definitions; the differences arise in the second term. Depending on the setting of $\Omega_y$, we can make the following observations:

- By definition we require that $j \in \Omega_y \subseteq [m] \setminus \{y\}$, thus $\max_{k \neq y} [h_\theta(\boldsymbol{x})]_k \geq \max_{k \in \Omega_y} [h_\theta(\boldsymbol{x})]_k \geq [h_\theta(\boldsymbol{x})]_j$. Due to the monotonicity of $\Phi^{-1}$, we therefore have $r(\boldsymbol{x}) \leq r_{\text{cs-group}}(\boldsymbol{x}; \Omega_y) \leq r_{\text{cs-pair}}(\boldsymbol{x}; j)$.

- When $|\Omega_y| = m - 1$, $\Omega_y = \{j | j \neq y, j \in [m]\}$ encompasses all incorrect classes, the two probability terms are fully matched for $r_{\text{cs-group}}(\boldsymbol{x}; \Omega_y)$ and $r(\boldsymbol{x})$, leading to $r_{\text{cs-group}}(\boldsymbol{x}; \Omega_y) = r(\boldsymbol{x})$.

- When $|\Omega_y| = 1$, specifically $\Omega_y = \{j\}$, it is straightforward to observe that $\max_{k \in \Omega_y} [h_\theta(\boldsymbol{x})]_k = [h_\theta(\boldsymbol{x})]_j$, resulting in $r_{\text{cs-group}}(\boldsymbol{x}; \Omega_y) = r_{\text{cs-pair}}(\boldsymbol{x}; j)$.

Therefore, we complete the proof of Theorem 4.2.   □

### B.2. Proof of Theorem 4.4

*Proof of Theorem 4.4.* Our goal is to show that $\hat{r}_{\text{cs}}$ (i.e., both $\hat{r}_{\text{cs-group}}$ and $\hat{r}_{\text{cs-pair}}$) specified in Algorithms 1 and 2 is a cost-sensitive certified radius with at least $(1 - \alpha)$ confidence over the randomness of the Gaussian sampling. Let $m$ be the total number of label classes, and let $(p_1, \ldots, p_m)$ be the ground-truth probability distribution of the smoothed classifier $g_\theta$ for a given example $(\boldsymbol{x}, y)$. Denote by $p_A, p_B$ the maximum probabilities in $[m]$ and in $\Omega_y$, respectively. According to the design of Algorithms 1 and 2, we can compute the empirical estimate of $p_k$ for any $k \in [m]$ based on $n$ Gaussian samples and the base classifier $f_\theta$. Let $(\hat{p}_1, \ldots, \hat{p}_m)$ be the corresponding empirical estimates,

then we immediately know

$$\hat{p}_k \sim \text{Binomial}(n, p_k), \text{ for any } k \in [m].$$

For $\hat{r}_{\text{std}}$, we follow the procedure in Cohen et al. (2019) to compute the $(1 - \alpha)$ lower confidence bound by first calculating the $(1 - \alpha)$ lower confidence bound on $p_A$ (denoted as $\underline{p_A}$). Then, given the fact that $p_B \leq 1 - p_A$, this leads to an upper confidence bound on $\overline{p_B} = 1 - \underline{p_A}$. However, for the computation of $\hat{r}_{\text{cs}}$, we need to compute both a lower confidence bound on $p_A$ and an upper confidence bound on $p_B$, which requires additional care to make the computation rigorous. In particular, we adapt the definition of standard certified radius (Theorem 1 in Cohen et al. (2019)) to cost-sensitive scenarios for deriving $\hat{r}_{\text{cs}}$. Based on the construction $\underline{p_A} = \text{LCB}[\text{count}[\hat{c}_A], n, 1 - \alpha/2]$, we have

$$\Pr\left[\underline{p_A} \leq p_A\right] \geq 1 - \frac{\alpha}{2}. \tag{6}$$

Therefore, the remaining task is to prove

$$\Pr\left[\overline{p_B} \geq p_B\right] \geq 1 - \frac{\alpha}{2}. \tag{7}$$

If the above inequalities hold true, we immediately know that by the union bound,

$$\Pr\left[\hat{r}_{\text{cs}} \leq r_{\text{cs}}(\boldsymbol{x}; \Omega_y)\right]$$
$$= \Pr\left[\frac{\sigma}{2}\left(\Phi^{-1}(\underline{p_A}) - \Phi^{-1}(\overline{p_B})\right)\right.$$
$$\left. \leq \frac{\sigma}{2}\left(\Phi^{-1}(p_A) - \Phi^{-1}(p_B)\right)\right]$$
$$\geq 1 - \left(\Pr\left[\underline{p_A} \geq p_A\right] + \Pr\left[\overline{p_B} \leq p_B\right]\right)$$
$$\geq 1 - \alpha.$$

For the pairwise cost-sensitive certified radius, based on Equation 7, we directly have by setting:

$$\overline{p_B} = \text{UCB}(\text{count}[j], n, 1 - \frac{\alpha}{2}). \tag{8}$$

For the groupwise cost-sensitive certified radius,

$$\overline{p_B} = \max\{\text{UCB}(\text{count}[k], n, 1 - \frac{\alpha}{2|\Omega_y|}) : k \in \Omega_y\},$$

as defined in Algorithm 1. $\Omega_y$ denotes the set of cost-sensitive target classes, and $p_B = \max_{k \in \Omega_y}\{p_k\}$ is used to define $r_{\text{cs-group}}(\boldsymbol{x}, \Omega_y)$. The challenge for proving Equation 7 lies in the fact that we do not know the top class within $\Omega_y$ which is different from the case of $\underline{p_A}$. Therefore, we resort to upper bounding the maximum over all the ground-truth class probabilities within $\Omega_y$. Based on the distribution of $\hat{p}_k$, we know for any $k \in \Omega_y$,

$$\Pr\left[\overline{p_k} \geq p_k\right] \geq 1 - \alpha/(2|\Omega_y|), \tag{9}$$

where $\overline{p_k}$ is defined as the $(1 - \alpha/(2|\Omega_y|))$ upper confidence bound computed using $\hat{p}_k$. We remark that the choice of $(1 - \alpha/(2|\Omega_y|))$ can in fact be varied for each $k \in \Omega_y$ and even optimized for obtaining tighter bounds, as long as the summation of the probabilities of bad event happening is at most $\alpha/2$. Here, we choose the same value of $1 - \alpha/(2|\Omega_y|)$ across different $k$ for simplicity, which already achieves reasonably good performance in our preliminary experiments. According to the union bound, we have

$$\Pr\left[\max_{k \in \Omega_y}\overline{p_k} \geq p_B\right] = \Pr\left[\max_{k \in \Omega_y}\overline{p_k} \geq \max_{k \in \Omega_y} p_k\right]$$
$$\geq 1 - \sum_{k \in \Omega_y}\Pr\left[\overline{p_k} \leq p_k\right]$$
$$\geq 1 - |\Omega_y| \cdot \alpha/(2|\Omega_y|) = 1 - \frac{\alpha}{2},$$

where the first inequality holds because of the union bound, which completes the proof. $\square$

### B.3. Demonstrations for Remarks 4.6

In Figure 4, we illustrate various scenarios involving the probability prediction scores of $g_\theta$ on a given sample $(\boldsymbol{x}, y)$ that can lead to different orderings of the estimated certified radius. The plot considers an example with five label classes, where $p_y$ denotes the true underlying prediction probability of the smoothed classifier $g_\theta$ on the ground-truth class $y$, and $p_{k1}$ through $p_{k4}$ represent the remaining non-ground-truth classes. The corresponding colored curves depict the observable distributions of these underlying values, obtained through Monte Carlo sampling, which follow binomial distributions. We focus on the most favorable cases for $\hat{r}_{\text{std}}$ (as otherwise, our cost-sensitive radius is always larger) by setting $\Omega_y = \{k1, k2, k3, k4\}$, which includes all non-ground-truth classes, for the groupwise radius $r_{\text{cs-group}}(\boldsymbol{x}; \Omega_y)$, and designating $j$ as the index of the runner-up class (here $j = k1$, as shown in the figure) for the pairwise radius $r_{\text{cs-pair}}(\boldsymbol{x}; j)$. The figure sequentially displays all typical cases from top to bottom. We note the following points associated with the demonstration:

- The estimation of $\underline{p_A}$ generated by $\hat{r}_{\text{std}}$ is closer to the true underlying $p_A$ (as it allows a larger lower region, i.e., $\alpha$ vs. $\alpha/2$), while the differences between various estimates of $\underline{p_A}$ diminish with more random noisy samples and as $p_A$ approaches 1.

- When the prediction probability mass is not concentrated on the top two classes (e.g., subplot (a)), our cost-sensitive radii tend to be larger due to their more precise estimation of $\overline{p_B}$. However, when the prediction probability mass is concentrated on the top two classes (e.g., subplots (b) and (c)), $\hat{r}_{\text{std}}$ will be more advantageous.

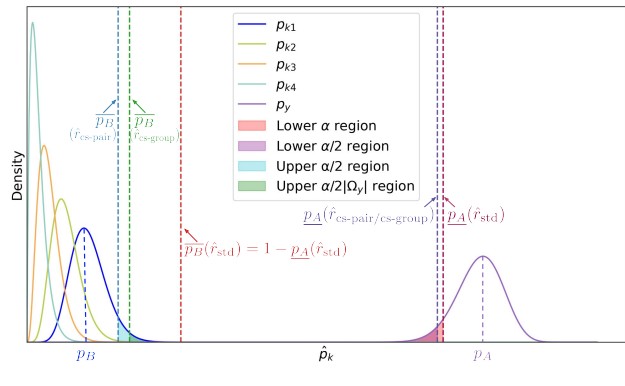

(a) $\hat{r}_{\text{std}} < \hat{r}_{\text{cs-group}} < \hat{r}_{\text{cs-pair}}$

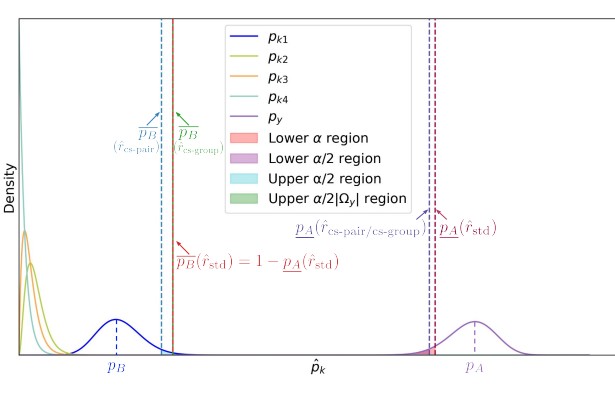

(b) $\hat{r}_{\text{cs-group}} < \hat{r}_{\text{std}} < \hat{r}_{\text{cs-pair}}$

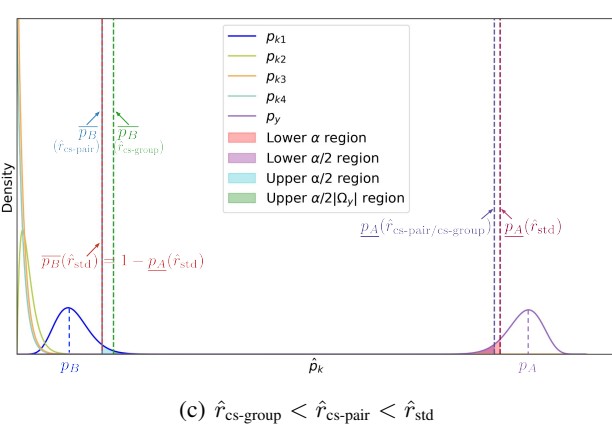

(c) $\hat{r}_{\text{cs-group}} < \hat{r}_{\text{cs-pair}} < \hat{r}_{\text{std}}$

*Figure 4.* Visualizations of scenarios that result in different orderings of the estimated certified radius. The vertical dashed lines, shown in different colors, indicate the lower and upper bounds for $p_A$ and $p_B$ as determined by different estimation methods.

- Comparing the two types of our cost-sensitive radius, $r_{\text{cs-pair}}$ will always be larger than $r_{\text{cs-group}}$, as they share the same estimation of $\underline{p_A}$, but the estimation of $\overline{p_B}$ from $r_{\text{cs-pair}}$ is closer to the true $p_B$ because it allows a larger upper confidence region (i.e., $\alpha/2$ vs. $\alpha/2|\Omega_y|$).

**Comparing $\hat{r}_{\text{cs}}$ with $\hat{r}_{\text{std}}$.** As noted in Remark 4.6, our certification algorithm tightens cost-sensitive robustness guar-

*Table 6.* Percentage for $\hat{r}_{\text{std}} > \hat{r}_{\text{cs-group}}$ under different cost matrices settings on CIFAR-10. We fix one seed class "cat" and vary the number of sensitive target classes $\Omega_y$.

| $|\Omega_y|$ | 1 | 3 | 5 | 7 | 9 |
|---|---|---|---|---|---|
| Gaussian | 0.9% | 1.7% | 13.7% | 16.2% | 22.1% |
| Margin-CS | 1.7% | 6.8% | 7.7% | 10.8% | 18.1% |

antees by returning the maximum of the standard estimate $\hat{r}_{\text{std}}$ and our cost-sensitive estimate $\hat{r}_{\text{cs}}$. To evaluate its effectiveness, we compare certified cost-sensitive robustness using $\hat{r}_{\text{std}}$ and $\hat{r}_{\text{cs-group}}$ under various settings. Figures 3(b) and 3(c) illustrate results for a fixed seed class "cat" (class "3") and cost-sensitive target sets of varying sizes. We assess models trained with Gaussian smoothing (Cohen et al., 2019) and our Margin-CS approach. Our results show that $\hat{r}_{\text{cs-group}}$ generally provides tighter cost-sensitive robustness guarantees than $\hat{r}_{\text{std}}$, with the gap increasing as $|\Omega_y|$ decreases. This occurs because a smaller $\Omega_y$ is less likely to include the runner-up class, causing the true $p_B$ (top class probability in $\Omega_y$) to be lower than the estimated $\overline{p_B}$ used for $\hat{r}_{\text{std}}$. These findings highlight the advantage of $\hat{r}_{\text{cs}}$ and the importance of our algorithm. However, as noted in Remark 4.6, when $\Omega_y$ includes the runner-up class, $\hat{r}_{\text{cs-group}}$ can occasionally be smaller than $\hat{r}_{\text{std}}$. Table 6 shows that this occurs in less than 20% of cases across various cost matrix settings. This supports our observation that $\hat{r}_{\text{std}}$ only outperforms $\hat{r}_{\text{cs-group}}$ when $\overline{p_B} = 1 - \underline{p_A}$ holds tightly, confirming the need for the maximum operation in our algorithm.

## C. Discussions & Insights

Existing robust training methods based on randomized smoothing can be roughly categorized into two main classes: (1) *noise-augmented training*, which enhance robustness by incorporating noise into the training process (Cohen et al., 2019; Salman et al., 2019; Jeong et al., 2021), and (2) *certified radius optimization*, which focus on directly maximizing the certified radius (Zhai et al., 2020). Both classes are considered in our work for completeness, as discussed in Section D.1 (in the appendix) and Section 5 (in the main paper), respectively. Below, we discuss the key strengths and limitations of each class in the context of cost-sensitive learning and compare them with alternative frameworks.

**Control vs. Flexibility.** Noise-augmented training methods are generally easy to implement and broadly applicable across various models and data modalities. However, this flexibility often comes at the cost of reduced granular control, making them less suitable for cost-sensitive scenarios. In such cases, adaptations typically rely on reweighting, a straightforward but restricted technique for cost-sensitive learning. Achieving more specific adaptations would require adjusting the noise distribution or perturbation strategy

for different cost matrices, which complicates implementation and yielded suboptimal results in our preliminary tests. While these methods generally offer reasonable performance, they often fall short of achieving optimal robustness. In contrast, certified radius optimization methods offer greater control by focusing directly on maximizing the robustness measure. This approach is well-suited for integrating the certified radius into cost-sensitive frameworks, allowing for more precise management of different misclassification types as defined by the cost matrix.

**Scalability vs. Tightness.** While this work focuses on the randomized smoothing framework, it is worth noting that the convex relaxation framework has also been adopted for cost-sensitive certification (Zhang & Evans, 2019). In comparison, this alternative framework may be slightly more advantageous in providing tighter robustness guarantees, as it allows for more precise control over the perturbation sets and leverages well-established optimization techniques such as semidefinite programming and duality theory. However, these advantages come at the expense of significant scalability challenges, rendering the approach infeasible for real-world, high-dimensional data. In contrast, randomized smoothing is highly scalable, making it well-suited for large datasets. However, our certification method, while smoothly scaling with input dimensions, faces a worst-case quadratic scaling with the number of label classes. This could be a concern when aiming for precise control over all types of misclassifications. Despite this, its practical impact is often mitigated in real-world scenarios, where strong priors exist for only a few critical misclassification types. In fact, many cost-sensitive learning tasks in practice are either binary or involve a small subset of classes where misclassification is particularly costly. This results in a sparse cost matrix, which simplifies the optimization process and ensures that our approach remains efficient in typical practical scenarios.

# D. Additional Details & Experiments Results

## D.1. Adaptive Baselines

The randomized-smoothing framework is built on the principle of introducing noise into input samples during inference. To further enhance both performance and robustness, existing works typically proposed accounting for this noise when updating the base or smoothed classifier (Cohen et al., 2019; Salman et al., 2019; Jeong et al., 2021) (i.e., noise-augmented training). In this context, adapting to cost-sensitive samples can be effectively addressed through *reweighting*, a standard and widely accepted approach in cost-sensitive learning. By assigning higher importance to these sensitive samples during the training process, reweighting ensures the model is better equipped to handle misclassifications that incur higher costs. In this section, we present

these (cost-sensitive) adaptive baselines.

**Gaussian-CS.** We first consider the base classifier training method introduced in Cohen et al. (2019), which proposes injecting Gaussian noise into all inputs during the training of $f_\theta$ to mitigate the negative effects of the noise introduced during inference. In cost-sensitive settings, the reweighting scheme involves increasing the weights assigned to the loss function of sensitive examples, a method we refer to as *Gaussian-CS*. Specifically, the training objective of *Gaussian-CS* is defined as follows:

$$
\min_{\theta \in \Theta} \Bigg[ \mathbb{E}_{(\boldsymbol{x},y)\sim\mathcal{D}_n} \mathbb{E}_{\boldsymbol{\delta}\sim\mathcal{N}(\mathbf{0},\sigma^2\mathbf{I})} \, \mathcal{L}_{\text{CE}}\big(f_\theta(\boldsymbol{x}+\boldsymbol{\delta}),y\big)
$$
$$
+ \lambda \cdot \mathbb{E}_{(\boldsymbol{x},y)\sim\mathcal{D}_s} \mathbb{E}_{\boldsymbol{\delta}\sim\mathcal{N}(\mathbf{0},\sigma^2\mathbf{I})} \, \mathcal{L}_{\text{CE}}\big(f_\theta(\boldsymbol{x}+\boldsymbol{\delta}),y\big) \Bigg],
$$

where $\lambda \geq 1$ is a trade-off parameter that controls the performance between sensitive and non-sensitive examples. When $\lambda = 1$, the above objective function is equivalent to the training loss used in standard randomized smoothing.

**SmoothAdv-CS.** *SmoothAdv* (Salman et al., 2019) applies adversarial training to the smoothed classifiers $g_\theta$ to improve its certified robustness, which involves incorporating adversarial samples into the updating process of the smoothed classifier. Similarly to *Gaussian-CS*, we adapt *SmoothAdv* by reweighting the cost-sensitive samples:

$$
\min_{\theta \in \Theta} \Bigg[ \mathbb{E}_{(\boldsymbol{x},y)\sim\mathcal{D}_n} \mathbb{E}_{\boldsymbol{\delta}\sim\mathcal{N}(\mathbf{0},\sigma^2\mathbf{I})} \, \mathcal{L}_{\text{CE}}\big(f_\theta(\tilde{\boldsymbol{x}}+\boldsymbol{\delta}),y\big)
$$
$$
+ \lambda \cdot \mathbb{E}_{(\boldsymbol{x},y)\sim\mathcal{D}_s} \mathbb{E}_{\boldsymbol{\delta}\sim\mathcal{N}(\mathbf{0},\sigma^2\mathbf{I})} \, \mathcal{L}_{\text{CE}}\big(f_\theta(\tilde{\boldsymbol{x}}+\boldsymbol{\delta}),y\big) \Bigg],
$$

where $\hat{\boldsymbol{x}}$ represents adversarial example of $\boldsymbol{x}$ against smoothed classifiers $g_\theta$, which is found by:

$$
\tilde{\boldsymbol{x}} = \underset{\boldsymbol{x}',\|\boldsymbol{x}'-\boldsymbol{x}\|_2 \leq \epsilon}{\arg\max} \, \mathcal{L}_{CE}(g_\theta(\boldsymbol{x}'),y)
$$
$$
\approx \underset{\boldsymbol{x}',\|\boldsymbol{x}'-\boldsymbol{x}\|_2 \leq \epsilon}{\arg\max} \, \mathcal{L}_{CE}(G_\theta(\boldsymbol{x}'),y) \tag{10}
$$
$$
= \underset{\boldsymbol{x}',\|\boldsymbol{x}'-\boldsymbol{x}\|_2 \leq \epsilon}{\arg\max} \Big( -\log \mathbb{E}_{\boldsymbol{\delta}\sim\mathcal{N}(\mathbf{0},\sigma^2 I)} \big[ F_\theta(\boldsymbol{x}'+\boldsymbol{\delta}) \big]_y \Big),
$$
$$
\tag{11}
$$

where $\epsilon$ denotes the maximum allowable $\ell_2$-norm distance between the adversarial example $\boldsymbol{x}'$ and the original input $\boldsymbol{x}$, $F_\theta$ and $G_\theta$ represent the soft-base and soft-smoothed classifiers, respectively. The approximation in Equations 10 and 11 introduces a differentiable objective, which is further approximated using Monte Carlo sampling with a small number of Gaussian noise samples for $\boldsymbol{\delta}$.

**SmoothMix-CS.** Smoothed classifiers exhibit a crucial relationship between prediction confidence and adversarial

robustness: higher confidence typically leads to better certified robustness. In this regard, *SmoothMix* (Jeong et al., 2021) suggests that the "over-confident but semantically off-class" samples can undermine the classifier's robustness. They further observe that such over-confident examples can be efficiently found along the direction of adversarial perturbations for a given input and propose to regularize the over-confident predictions along the adversarial direction toward the uniform prediction through a mixup loss. The training objective is as follows:

$$\min_{\theta \in \Theta} \left[ \mathbb{E}_{(\boldsymbol{x},y) \sim \mathcal{D}} \mathbb{E}_{\boldsymbol{\delta} \sim \mathcal{N}(\boldsymbol{0},\sigma^2 \mathbf{I})} \, \mathcal{L}_{\mathrm{CE}}\big(f_\theta(\boldsymbol{x}+\boldsymbol{\delta}), y\big) \right. \tag{12}$$

$$\left. + \eta \cdot \mathbb{E}_{(\boldsymbol{x},y) \sim \mathcal{D}} \mathbb{E}_{\boldsymbol{\delta} \sim \mathcal{N}(\boldsymbol{0},\sigma^2 \mathbf{I})} \, \mathcal{L}_{\mathrm{CE}}\big(f_\theta(\boldsymbol{x}^{\mathrm{mix}}+\boldsymbol{\delta}), y^{\mathrm{mix}}\big) \right],$$

$$\text{where} \quad \boldsymbol{x}^{\mathrm{mix}} = (1-t) \cdot \boldsymbol{x} + t \cdot \tilde{\boldsymbol{x}}',$$

$$y^{\mathrm{mix}} = (1-t) \cdot \widehat{G}_\theta(\boldsymbol{x}) + t \cdot \frac{1}{m},$$

$$t \sim U([0, 1/2]),$$

Here, $t$ is uniformly sampled from $[0, 1/2]$, forming a convex combination between the original input and an adversarial sample, $\frac{1}{m}$ is the uniform probability vector over the $m$ labels, and $\widehat{G}_\theta(\boldsymbol{x})$ denotes the estimated soft-smoothed prediction of $\boldsymbol{x}$, calculated as the average softmax predictions of the base classifier on the noisy sample $\boldsymbol{x} + \boldsymbol{\delta}$. The $\tilde{\boldsymbol{x}}'$ corresponds to the unrestricted adversarial sample of the smoothed classifier $g_\theta$ for $\boldsymbol{x}$, which is obtained by solving:

$$\tilde{\boldsymbol{x}}' = \arg\max_{\boldsymbol{x}'} \left( \mathcal{L}_{CE}(g_\theta(\boldsymbol{x}'), y) - \beta \|\boldsymbol{x}' - \boldsymbol{x}\|_2^2 \right). \tag{13}$$

Here, $\beta > 0$ ensures that the adversarial sample $\tilde{\boldsymbol{x}}'$ remains within a reasonable distance from $\boldsymbol{x}$, preventing it from being arbitrarily far. Note that while both $\boldsymbol{x}^{\mathrm{mix}}$ and $y^{\mathrm{mix}}$ depends on the model parameter $\theta$, they are used only as targets during the forward pass, and the gradient with respect to $\theta$ from this path is not propagated in the backward pass. For our reweighting adaptation, we modify the first term of the objective in Equation 12 to rebalance the robustness of cost-sensitive examples and normal examples. The final objective is defined as:

$$\min_{\theta \in \Theta} \left[ \mathbb{E}_{(\boldsymbol{x},y) \sim \mathcal{D}_n} \mathbb{E}_{\boldsymbol{\delta} \sim \mathcal{N}(0,\sigma^2 \mathbf{I})} \, \mathcal{L}_{\mathrm{CE}}\big(f_\theta(\boldsymbol{x}+\boldsymbol{\delta}), y\big) \right.$$

$$+ \lambda \cdot \mathbb{E}_{(\boldsymbol{x},y) \sim \mathcal{D}_s} \mathbb{E}_{\boldsymbol{\delta} \sim \mathcal{N}(0,\sigma^2 \mathbf{I})} \, \mathcal{L}_{\mathrm{CE}}\big(f_\theta(\boldsymbol{x}+\boldsymbol{\delta}), y\big) \tag{14}$$

$$\left. + \eta \cdot \mathbb{E}_{(\boldsymbol{x},y) \sim \mathcal{D}} \mathbb{E}_{\boldsymbol{\delta} \sim \mathcal{N}(0,\sigma^2 \mathbf{I})} \, \mathcal{L}_{\mathrm{CE}}\big(f_\theta(\boldsymbol{x}^{\mathrm{mix}}+\boldsymbol{\delta}), y^{\mathrm{mix}}\big) \right].$$

### D.2. Datasets

**CIFAR-10.** The CIFAR-10 dataset[2] consists of 60,000

$32 \times 32$ colour images in 10 classes (airplane, automobile, bird, cat, deer, dog, frog, horse, ship, truck), with 6,000 images per class, with 50,000 training images and 10,000 test images. The training set contains exactly 5000 images per class, while the test set contains exactly 1,000 randomly selected images per class, making it a balanced dataset.

**Imagenette.** The Imagenette dataset[3] is a curated subset of 10 easily recognizable classes from the larger ImageNet dataset. It includes images from the following categories: tench, English springer, cassette player, chainsaw, church, French horn, garbage truck, gas pump, golf ball, and parachute. The images are resized to $160 \times 160$ pixels, making it a compact and focused dataset ideal for efficient experimentation and model benchmarking.

**ImageNet.** ImageNet (Russakovsky et al., 2015) is a large-scale visual database widely used in visual object recognition research. It contains over 14 million color images spanning more than 20,000 categories. A popular subset of ImageNet, commonly used in research, comprises 1.2 million high-resolution images categorized into 1,000 classes, including animals, vehicles, and everyday objects. The dataset is divided into a training set with 1.2 million images and a validation set with 50,000 images. The validation set includes 50 images per class, ensuring balanced representation across all categories. During preprocessing, the images are typically cropped to a size of $224 \times 224$ pixels.

**HAM10k.** The HAM10k dataset (Tschandl et al., 2018) comprises 10,015 images resulting from a comprehensive study conducted by multiple entities. Each image is in RGB format and has dimensions of $600 \times 450$ pixels (length $\times$ width). During preprocessing, the images are typically cropped to a size of $299 \times 299$ pixels. The dataset is designed to facilitate the study and classification of seven distinct types of skin lesions, including "Melanocytic nevi" (6705 samples), "Dermatofibroma" (1113 samples), "Benign keratosis-like lesions" (1099 samples), "Basal cell carcinoma" (514 samples), "Actinic keratoses" (327 samples), "Vascular lesions" (142 samples), and "Dermatofibroma" (115 samples). Among these types, "Dermatofibroma" and "Basal cell carcinoma" are malignant, while the others are benign, forming a highly imbalanced distribution between malignant and benign samples, closely mirroring real-world conditions. This collection serves as a valuable resource for advancing research in dermatological image analysis and skin lesion classification.

### D.3. Hyperparameters

For Gaussian-CS, SmoothAdv-CS and SmoothMix-CS, the parameter $\lambda$ is carefully tuned to ensure the best trade-off be-

---

[2]https://www.cs.toronto.edu/ kriz/cifar.html

[3]https://github.com/fastai/imagenette

*Table 7.* Performance metrics under different $(\lambda_1, \lambda_2)$ settings.

| $\lambda_1$ | $\lambda_2$ | $\mathbf{Acc} \uparrow$ | $\mathbf{Rob_{cs}} \uparrow$ | $\mathbf{Rob_{cost}} \downarrow$ |
|---|---|---|---|---|
| 1 | 1 | 0.69 | 0.22 | 5.15 |
| 2 | 2 | 0.68 | 0.45 | 3.62 |
| 3 | 3 | 0.67 | 0.51 | 3.41 |
| 4 | 4 | 0.63 | 0.73 | 1.60 |
| 5 | 5 | 0.60 | 0.76 | 1.37 |
| 6 | 6 | 0.58 | 0.81 | 1.05 |

tween overall accuracy and cost-sensitive robustness, where we enumerate all values from $\{1.0, 1.1, \ldots, 2.0\}$ and observe nearly in all cases of cost matrices, setting $\lambda = 1.1$ achieves the best result. For MACER, the parameter $\lambda$ (which corresponds to the ratio between cross-entropy loss and robustness loss in Zhai et al. (2020) is fixed at 4 by default. Similarly, in our Margin-CS method, we set $\lambda_1 = 3$ and $\lambda_2 = 3$ according to observation from Table 7.

We present the results of varying hyperparameters $\gamma_1$ and $\gamma_2$ in our Margin-CS method, evaluated across the main metrics: $\mathbf{Acc}$, $\mathbf{Rob_{cs}}$ and $\mathbf{Rob_{cost}}$. Our goal is to improve cost-sensitive robustness without sacrificing overall accuracy, where $\gamma_1$ controls the margin for normal classes and $\gamma_2$ for sensitive classes. We report results on CIFAR-10 under the "S-Seed" setting, where "cat" is selected as the sensitive seed class and misclassifications to other classes incur equal cost. This choice serves illustrative purposes, as similar trends are observed across other cost matrices, consistent with the results in Tables 1 and 2. A grid search (Table 8) reveals that the combination $(\gamma_1, \gamma_2) = (4, 16)$ yields satisfactory results.

### D.4. Detailed Experimental Setup

In our experiments, we use $\hat{r}_{std}, \hat{r}_{cs\text{-}group}, \hat{r}_{cs\text{-}pair}$ returned by Algorithms 1 and 2 as the empirical approximation of the true certified radius defined in Lemma 3.2 and Definition 4.1. For all cost-sensitive learning methods, we choose the hyperparameters based on the following selection criteria: it should yield a high $\mathbf{Rob_{cs}}$ (or low $\mathbf{Rob_{cost}}$) while $\mathbf{Acc}$ should be comparable to that of the baseline randomized smoothing methods designed for maximizing overall robustness.

For CIFAR-10, Imagenette, and HAM10k, each experiment is run on a single NVIDIA A100 GPU with 40 GB of memory within one day. For the ImageNet dataset, each experiment is conducted on four NVIDIA A100 GPUs with 40 GB of memory for 1-2 days.

For the experiments with *Equal Costs for Selected Misclassifications* (Tables 1 and 2 in the main paper), we use third class "cat" (label 3) for CIFAR-10 and "gas pump" (label 7) for Imagenette in the *"S-Seed"* case. For the *"M-Seed"* case, "bird" (label 2) and "deer" (label 4) are considered as the sen-

sitive seed classes for CIFAR-10, while we choose "chain saw" (label 3) and "gas pump" (label 7) for Imagenette. For the *"S-Pair"* setting in CIFAR-10, we examine the misclassification of "cat" (label 3) to "dog" (label 5), a common and challenging confusion given their visual similarity. The corresponding *"M-Pair"* setting addresses misclassifications of "cat" (label 3) to "bird" (label 2), "deer" (label 4), and "dog" (label 5), covering a broader range of potential errors across both similar and dissimilar classes. In Imagenette, the *"S-Pair"* setting focuses on the misclassification of "gas pump" (label 7) to "church" (label 2), highlighting a scenario where structural similarities can lead to errors. The *"M-Pair"* setting for Imagenette considers misclassifications of "gas pump" (label 7) to "church" (label 2), "chain saw" (label 4), and "garbage truck" (label 6), addressing a diverse set of challenging misclassifications that involve both contextual and structural similarities.

### D.5. Certification Results with Varying Noises.

**Results with Varying $\sigma$.** We test the performance of our method under fixed $\ell_2$ perturbations with $\epsilon = 0.25$ when the standard deviation parameter of the injected Gaussian noises is $\sigma = 0.25$. We demonstrate the results in Tables 9 and 10. The results show that our method consistently outperforms several baseline randomized smoothing methods and their reweighting counterparts.

**Results with Varying $\epsilon$.** We test the performance of our method under various $\ell_2$ perturbations with different $\epsilon$ values and fix the standard deviation parameter of the injected Gaussian noise to $\sigma = 0.5$, same as in Section 6. The results, shown in Figure 5 for both seedwise and pairwise cost matrices, demonstrate the superiority of our method compared to several baseline randomized smoothing methods and their reweighting counterparts.

### D.6. Scalability for Large Datasets

We additionally validate the scalability of our certification framework and training methods by applying them to the full ImageNet dataset, where most traditional certification techniques fall short. The results, presented in Table 11, cover three typical scenarios involving selected seed classes with equal costs assigned to misclassification into all other classes. The *"S-Seed"* scenario focuses on class labels "919" and "920", representing "street sign" and "traffic light and signals", respectively. Additionally, the *"M-Seeds"* scenario includes class labels "44" and "48", corresponding to different types of giant lizards. Misclassifications in these ImageNet classes can have serious consequences, such as safety risks in autonomous driving or real-world dangers.

Table 11 shows that our methods effectively enhance cost-sensitive robustness by approximately 15%-20% in $\mathbf{Rob_{cs}}$

*Table 8.* Performance of Margin-CS across different combinations of $\gamma_1$ and $\gamma_2$ on CIFAR-10 (*"S-Seed"*).

| | $\gamma_2 = 8$ | | | $\gamma_2 = 10$ | | | $\gamma_2 = 12$ | | | $\gamma_2 = 16$ | | |
|---|---|---|---|---|---|---|---|---|---|---|---|---|
| | Acc ↑ | Rob$_{cs}$ ↑ | Rob$_{cost}$ ↓ | Acc ↑ | Rob$_{cs}$ ↑ | Rob$_{cost}$ ↓ | Acc ↑ | Rob$_{cs}$ ↑ | Rob$_{cost}$ ↓ | Acc ↑ | Rob$_{cs}$ ↑ | Rob$_{cost}$ ↓ |
| $\gamma_1 = 2$ | 65.4 | 63.3 | 2.62 | 63.4 | 68.7 | 2.49 | 63.7 | 69.1 | 2.48 | 63.0 | 70.5 | 2.51 |
| $\gamma_1 = 4$ | 67.0 | 50.7 | 3.56 | 65.3 | 59.7 | 3.48 | 65.9 | 57.6 | 3.44 | 66.1 | 58.3 | 3.04 |
| $\gamma_1 = 6$ | 67.3 | 39.6 | 3.88 | 66.0 | 49.3 | 3.55 | 65.5 | 54.4 | 3.36 | 64.9 | 55.2 | 3.23 |
| $\gamma_1 = 8$ | 66.0 | 33.8 | 4.39 | 65.0 | 43.2 | 4.14 | 64.1 | 47.4 | 3.92 | 64.5 | 46.3 | 3.77 |

*Table 9.* Certification results on CIFAR-10 for selected misclassifications with equal costs ($\sigma = \epsilon = 0.25$).

| Method | S-Seed | | | M-Seed | | | S-Pair | | | M-Pair | | |
|---|---|---|---|---|---|---|---|---|---|---|---|---|
| | Acc ↑ | Rob$_{cs}$ ↑ | Rob$_{cost}$ ↓ | Acc ↑ | Rob$_{cs}$ ↑ | Rob$_{cost}$ ↓ | Acc ↑ | Rob$_{cs}$ ↑ | Rob$_{cost}$ ↓ | Acc ↑ | Rob$_{cs}$ ↑ | Rob$_{cost}$ ↓ |
| Gaussian | 79.3 | 40.7 | 3.79 | 79.3 | 58.8 | 2.96 | 79.3 | 67.4 | 0.56 | 79.3 | 51.5 | 1.27 |
| SmoothMix | 73.0 | 32.6 | 3.99 | 73.0 | 46.4 | 3.63 | 73.0 | 72.0 | 0.31 | 73.0 | 56.3 | 1.16 |
| SmoothAdv | 77.9 | 42.8 | 3.66 | 77.9 | 52.5 | 2.89 | 77.9 | 73.8 | 0.32 | 77.9 | 56.0 | 1.07 |
| MACER | **80.8** | 47.5 | 3.30 | **80.8** | 65.3 | 2.72 | 80.8 | 70.9 | 0.34 | 80.8 | 58.2 | 1.06 |
| Gaussian-CS | 77.8 | 58.3 | 2.39 | 78.4 | 63.5 | 2.61 | 77.8 | 81.1 | 0.39 | 77.8 | 68.5 | 1.18 |
| SmoothMix-CS | 72.2 | 60.2 | 2.99 | 72.1 | 47.3 | 2.85 | 72.2 | 75.7 | 0.26 | 72.2 | 73.8 | 0.70 |
| SmoothAdv-CS | 78.7 | 62.5 | 2.47 | 78.4 | 56.6 | 2.65 | 78.7 | 77.4 | 0.28 | 78.1 | 66.3 | 1.03 |
| Margin-CS | 80.4 | **68.8** | **0.78** | 80.7 | **71.7** | **2.39** | 80.9 | **93.5** | **0.19** | 80.9 | **91.4** | **0.24** |

*Table 10.* Certification results on Imagenette for selected misclassifications with equal costs ($\sigma = \epsilon = 0.25$).

| Method | S-Seed | | | M-Seed | | | S-Pair | | | M-Pair | | |
|---|---|---|---|---|---|---|---|---|---|---|---|---|
| | Acc ↑ | Rob$_{cs}$ ↑ | Rob$_{cost}$ ↓ | Acc ↑ | Rob$_{cs}$ ↑ | Rob$_{cost}$ ↓ | Acc ↑ | Rob$_{cs}$ ↑ | Rob$_{cost}$ ↓ | Acc ↑ | Rob$_{cs}$ ↑ | Rob$_{cost}$ ↓ |
| Gaussian | 80.3 | 65.6 | 3.41 | 80.3 | 61.2 | 3.21 | 80.3 | 88.0 | 0.22 | 80.3 | 73.0 | 1.20 |
| SmoothMix | 81.4 | 61.9 | 3.06 | 81.4 | 59.2 | 2.82 | 81.4 | 83.3 | 0.25 | 81.4 | 78.5 | 1.71 |
| SmoothAdv | 77.8 | 66.6 | 3.30 | 77.8 | 59.3 | 3.08 | 77.8 | 90.9 | 0.20 | 77.8 | 80.4 | 1.67 |
| MACER | 79.6 | 70.1 | 2.78 | 79.6 | 66.2 | 3.13 | 79.6 | 83.7 | 0.21 | 79.6 | 78.0 | 1.54 |
| Gaussian-CS | 73.8 | 74.2 | 2.64 | 76.4 | 65.6 | 2.83 | 76.4 | 92.1 | 0.17 | 76.4 | 83.1 | 1.07 |
| SmoothMix-CS | 78.1 | 71.4 | 2.81 | 79.2 | 70.7 | 2.76 | 78.1 | 84.8 | 0.11 | 78.1 | 81.8 | 1.69 |
| SmoothAdv-CS | 77.7 | 70.6 | 2.91 | 77.7 | 60.5 | 2.74 | 77.7 | 91.8 | 0.08 | 77.6 | 82.0 | 1.06 |
| Margin-CS | **83.6** | **75.6** | **2.34** | **81.9** | **73.2** | **2.52** | **83.6** | **94.2** | **0.04** | **82.6** | **88.0** | **0.84** |

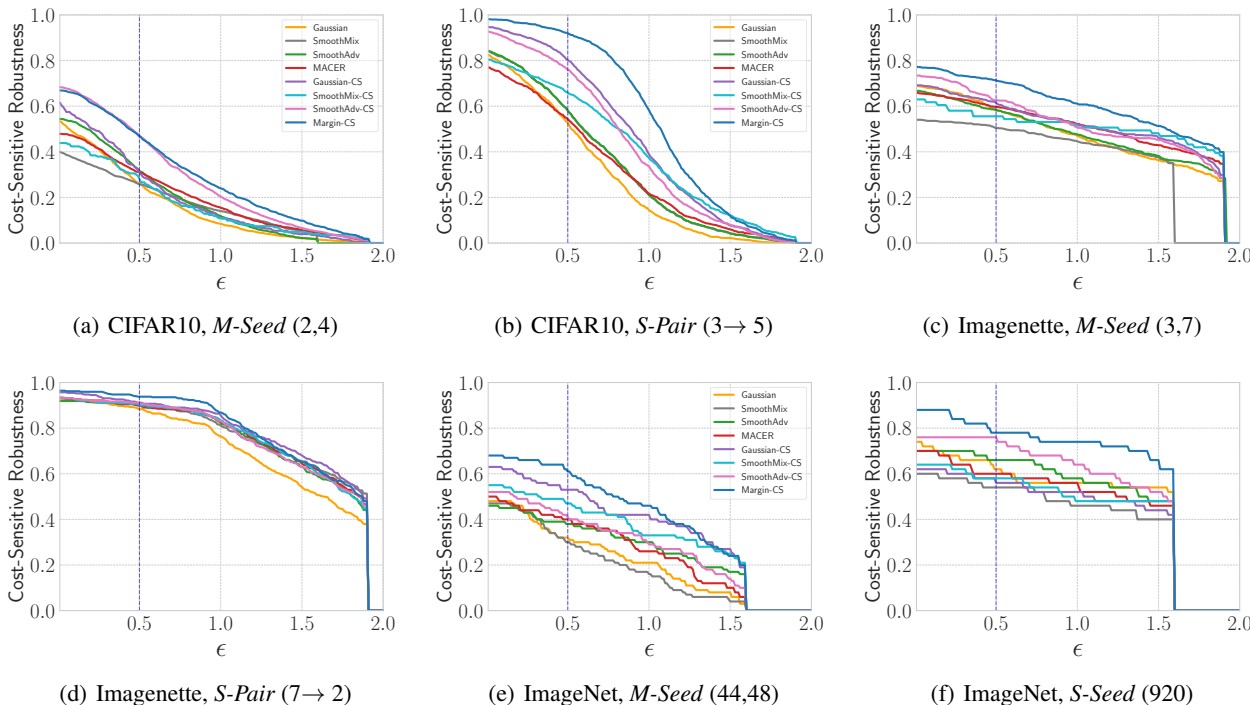

(a) CIFAR10, *M-Seed* (2,4)     (b) CIFAR10, *S-Pair* (3→5)     (c) Imagenette, *M-Seed* (3,7)

(d) Imagenette, *S-Pair* (7→2)     (e) ImageNet, *M-Seed* (44,48)     (f) ImageNet, *S-Seed* (920)

*Figure 5.* Certified accuracy curves for varying $\epsilon$ for different datasets and various cost matrix settings.

*Table 11.* Results (in %) across various cost matrices on ImageNet.

| Method | M-Seed (44, 48) | | S-Seed (919) | | S-Seed (920) | |
|---|---|---|---|---|---|---|
| | Acc ↑ | Rob$_{cs}$ ↑ | Acc ↑ | Rob$_{cs}$ ↑ | Acc ↑ | Rob$_{cs}$ ↑ |
| Gaussian | 58.4 | 45.0 | 58.4 | 58.0 | 58.4 | 52.0 |
| SmoothMix | 54.4 | 41.4 | 54.4 | 52.0 | 54.4 | 54.0 |
| SmoothAdv | 57.1 | 38.0 | 57.1 | 70.0 | 57.1 | 67.0 |
| MACER | **58.5** | 40.0 | **58.5** | 66.0 | **58.5** | 62.0 |
| Gaussian-CS | 48.6 | 53.0 | 52.4 | 78.0 | 51.0 | 62.0 |
| SmoothMix-CS | 48.6 | 47.0 | 51.6 | 70.0 | 51.2 | 58.0 |
| SmoothAdv-CS | 50.4 | 41.0 | 50.6 | 76.0 | 52.4 | **78.0** |
| Margin-CS | 54.2 | **61.0** | 53.4 | **84.0** | 53.6 | 76.0 |

Margin-CS method is generally concentrated at larger values, which aligns with the quantitative results in Tables 1 and 2, demonstrating that Margin-CS generally achieves best certifiable robustness among all methods. Moreover, the improvement in certified radius is less pronounced for normal (i.e., non-sensitive) samples compared to sensitive samples, aligning with our primary goal of cost-sensitive learning, which prioritizes the accurate classification of "sensitive" samples with high practical value.

compared to their standard counterparts across various cost-matrix settings, while largely preserving utility. Margin-CS consistently achieves the best cost-sensitive robustness, which aligns with our observations on other standard benchmark datasets presented in the aforementioned sections.

### D.7. Heatmap Analysis

To further analyze the trade-offs between our Margin-CS method and its non-cost-sensitive counterpart MACER, we present heatmaps illustrating clean and robust test errors in Figure 6. Compared to MACER, our Margin-CS method tends to focus more on the sensitive seed class "cat", significantly reducing misclassifications to other classes. This results in a lower overall clean test error for our method compared to MACER. Specifically, for the sensitive seed class "cat", the clean test error decreases from 40.1% to 11.1%, demonstrating the effectiveness of our cost-sensitive training approach.

Among the normal classes, the most notable impact of our method is on the "dog" class, where misclassification rates to "cat" increase from 13.6% to 44.3%, while misclassification rates to other classes decrease from 17.4% to 9.2%. In our *S-Seed* cost-matrix settings, this increase in misclassification for the "dog" class is acceptable, as long as the overall misclassification rates are controlled. A similar trend is observed in the robust test error map, where the robust test error for the "cat" class decreases, while that for the "dog" class increases. Along this line, an interesting direction for future work would be to develop more balanced and robust training methods.

### D.8. Visualization of Radius Distributions

We plot the distributions of certified radius provided by different robust training methods on the sensitive testing examples ($\mathcal{D}_s$) in Figure 7, the overall dataset ($\mathcal{D}$) in Figure 8, and the normal subset ($\mathcal{D}_n$) in Figure 9, respectively. As shown by comparing the two rows of subplots, all our cost-sensitive adaptive robust training methods shift the radius distributions towards larger values, indicating higher cost-sensitive robustness compared to their non-cost-sensitive counterparts. Additionally, the distribution produced by our

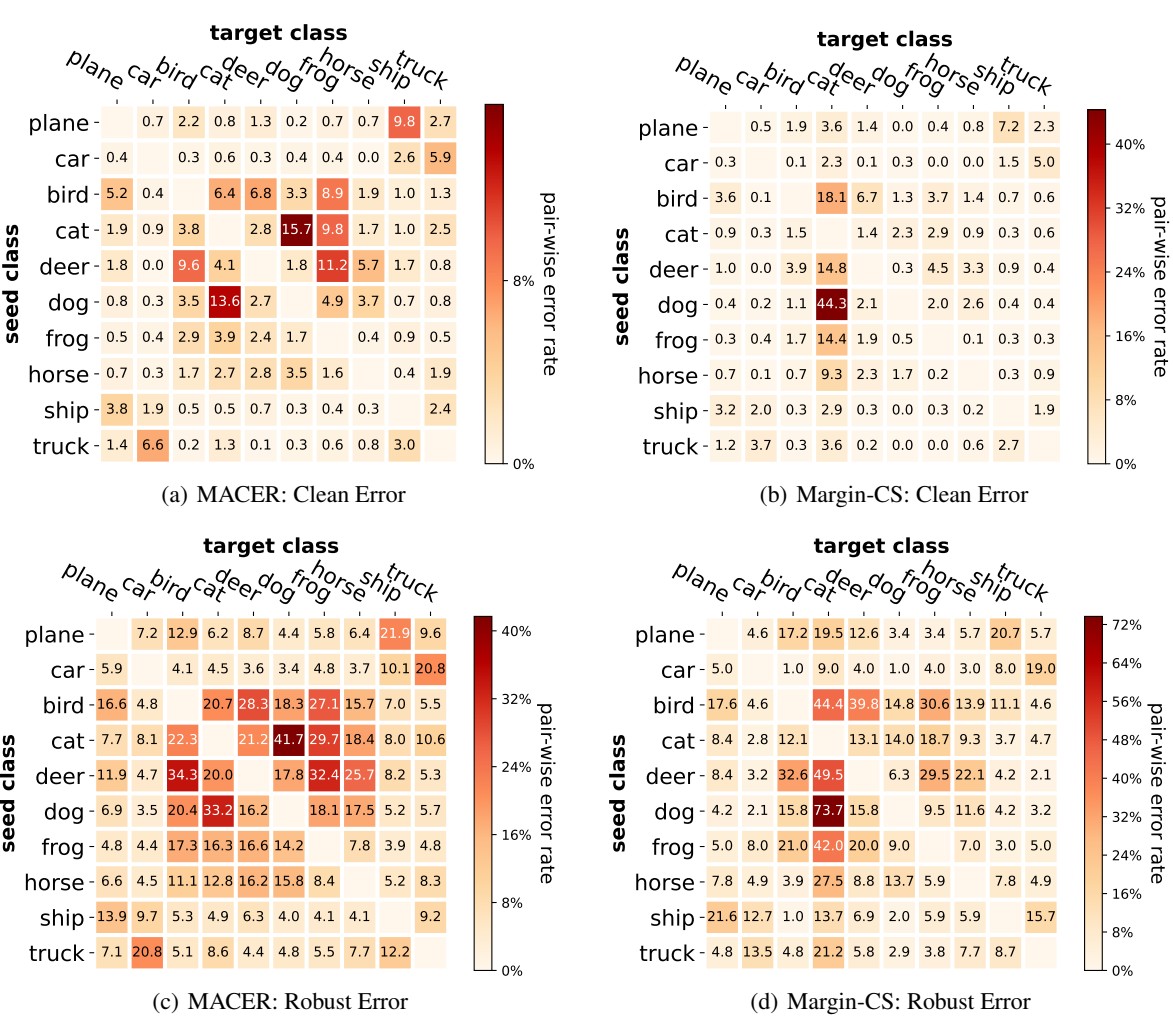

Figure 6. Figures **(a)** and **(b)**: Heatmaps of clean test error on CIFAR-10 between MACER and Margin-CS. Figures **(c)** and **(d)**: Heatmaps of robust test error on CIFAR-10 between MACER and Margin-CS. The cost matrix is set to be the *"S-Seed"* case.

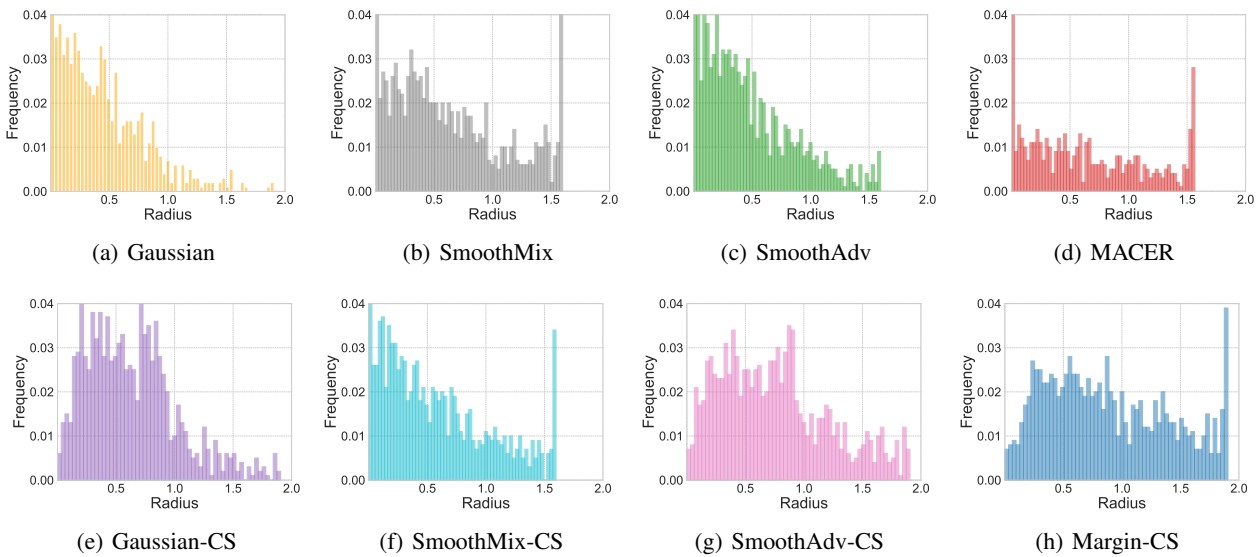

*Figure 7.* Distribution of $\hat{r}_{\text{cs-group}}$ on testing samples from $\mathcal{D}_s$ under *"S-Seed"* setting on CIFAR-10. The x-axes are uniformly scaled across all plots, and the y-axes are truncated at a frequency of $0.04$ for clarity in visualization.

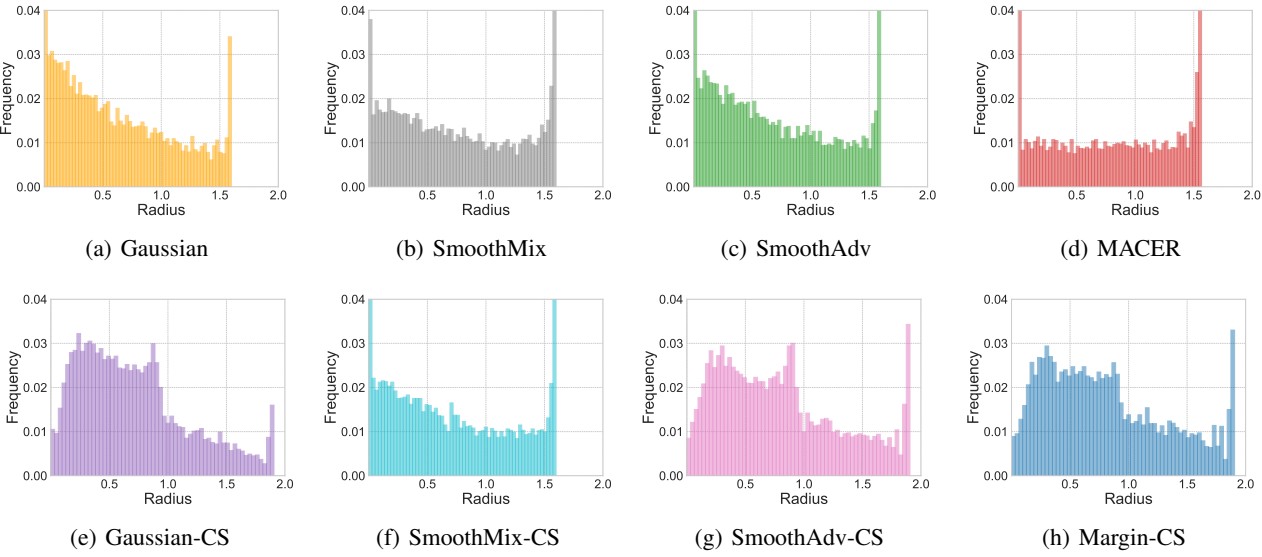

*Figure 8.* Distribution of $\hat{r}_{\text{cs-group}}$ on testing samples from $\mathcal{D}$ under *"S-Seed"* setting on CIFAR10. The x-axes are uniformly scaled across all plots, and the y-axes are truncated at a frequency of $0.04$ for clarity in visualization.

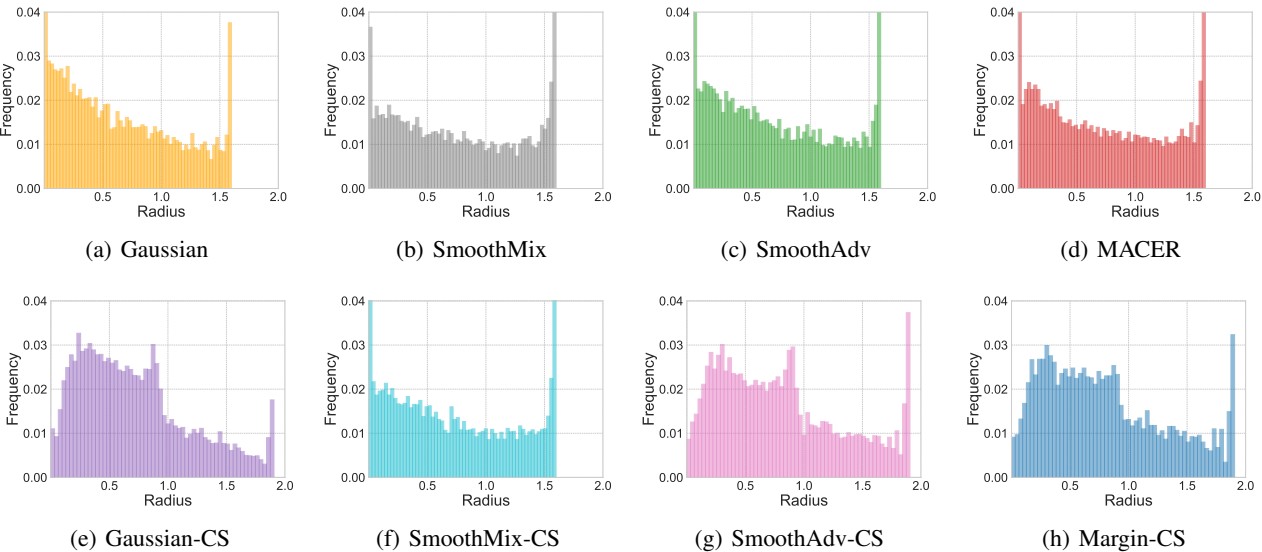

*Figure 9.* Distribution of $\hat{r}_{\text{cs-group}}$ on testing samples from $\mathcal{D}_n$ under *"S-Seed"* setting on CIFAR10. The x-axes are uniformly scaled across all plots, and the y-axes are truncated at a frequency of $0.04$ for clarity in visualization.

