# OpenReview forum: "Provably Cost-Sensitive Adversarial Defense via Randomized Smoothing"
_ICML.cc/2025/Conference — ICML 2025 poster_

### Official Review · Reviewer_KFXn · 2025-03-08

**Overall Recommendation:** 1

**Summary:**

This paper introduces an adversarial training algorithm aimed at enhancing cost-sensitive robustness. The writing is clear, and the methodology appears sound within the context of the paper. However, I have concerns regarding the motivation, and the evaluation lacks several critical experiments.

## Update after Rebuttal
The two-round rebuttal did not address my concerns, so I maintain my recommendation: 1. Reject.

There have been prior works on certified adversarial defense via random smoothing, as well as on cost-sensitive learning. My understanding of this work, after reading the rebuttal, is that it applies certified adversarial defense via random smoothing to cost-sensitive learning. This leads to my first and main concern: why this particular combination? Why not apply adversarial training or other defenses to cost-sensitive learning? Which method is more effective in this context?

The authors explain why randomized smoothing cannot be used for adversarial training, but that was not my question. What I asked is for a comparison between "adversarial training" and "certified adversarial defense via random smoothing" when applied to cost-sensitive learning. This key question remains unaddressed.

The reason I ask for a comparison (including results using standard classification metrics) is because the authors claim this work as an adversarial defense. A defense method must be evaluated under realistic threat models, considering the knowledge available to both attackers and defenders. However, the authors also acknowledge that certified defenses struggle against unseen attacks. On the other hand, although they claim that adversarial training generalizes poorly to unseen attacks, latest adversarial training was specifically designed to address this challenge and has evolved with many strategies to improve robustness (as shown in recent works listed in RobustBench). To support the claim that this method is practical and effective, the authors must provide comprehensive evaluations.

Even if we narrow the scope to the context of this study (applying certified adversarial defense via random smoothing to cost-sensitive learning), I still do not see references to or use of the latest methods in either area, even after the rebuttal. It is unclear whether the authors mean that there is no recent work on certified adversarial defense via random smoothing or cost-sensitive learning individually, or that there is no recent work combining the two.

My suggestion to the authors: If this work aims to bridge the gap between certified adversarial defense and cost-sensitive learning, then the narrative should not center on defense. Instead, the focus should be on understanding and addressing the gap between cost-sensitive and cost-insensitive learning when applying certified defenses. It is also important to discuss how different cost-sensitive learning algorithms and certified defense techniques may affect the effectiveness of bridging this gap.

**Claims And Evidence:**

* I do not agree with the motivation that defending against cost-sensitive adversaries is fundamentally different from regular adversarial training. The paper lacks both theoretical justification and empirical demonstrations to support this distinction.
* The paper does not report clean accuracy, despite the fact that maintaining clean accuracy in adversarial training is a well-known challenge. The authors claim in 033 that their method mitigates this issue, yet no supporting evidence is provided.
* It is essential to report the robust accuracy of both cost-sensitive and non-sensitive adversarial examples separately. The drop in robust accuracy for non-sensitive adversarial examples, similar to clean accuracy, should be carefully examined.
* The paper provides limited background on cost-sensitive robustness in introduction (only 050), which is crucial for distinguishing it from standard adversarial robustness. Readers must refer to Section 4 for further details.

**Essential References Not Discussed:**

The references are outdated, with no publications from 2023-2024 cited. Furthermore, the authors should include comparisons with regular adversarial training methods featured in RobustBench (https://github.com/RobustBench/robustbench).

**Experimental Designs Or Analyses:**

* The details of the attacks used in the experiments are not disclosed,
* The paper does not compare its approach with the most relevant work, such as Zhang et al. (2023), nor does it include comparisons with other regular adversarial training methods. Specifically, the overall accuracy presented in Table 1 and Table 11 is noticeably lower than that of state-of-the-art (SOTA) adversarial training methods (refer to RobustBenck).
* Given the lack of disclosure regarding attack details, it remains unclear whether the defense has been evaluated against unseen or adaptive attacks, which are crucial for assessing defenses.

**Methods And Evaluation Criteria:**

* The proposed methodology appears to be derived from Zhang et al. (2023), although their work was not specifically designed for cost-sensitive robustness. Could the authors clarify the differences in the algorithm, aside from the application/scenario context?

**Other Comments Or Suggestions:**

No comment.

**Other Strengths And Weaknesses:**

No comment.

**Questions For Authors:**

My current assessment falls between scores 1 and 2 due to several concerns. However, if these concerns are adequately addressed, I would be open to revising my score and potentially raising it above 2.
1. The proposed method should be compared with regular adversarial training methods, as these are designed to generalize across all adversarial scenarios. It would represent a good contribution if the proposed method demonstrates superior performance, even if only in the context of cost-sensitive robustness.
2. Clean accuracy, as well as robust accuracy for both cost-sensitive and non-sensitive adversarial examples, must be reported separately for a more comprehensive evaluation.
3. The novelty of the proposed method in comparison to Zhang et al. (2023) should be clearly articulated, beyond just the application domain.
4. The details of the attacks used in the experiments must be disclosed to ensure transparency and reproducibility.
5. A literature review that includes relevant works from 2024 is expected. If there are no significant new contributions, this should be explicitly stated for the readers.

**Relation To Broader Scientific Literature:**

No comment.

**Theoretical Claims:**

The authors provide sufficient mathematical proofs in the appendix.

---

> ### Author Rebuttal · Authors · 2025-03-31
>
> **1. It is essential to report the robust accuracy of both cost-sensitive and non-sensitive adversarial examples separately.**
>
> |Method|$Rob_{cs}$|$Rob_{normal}$|
> | -------- | -------- | --------
> |Gaussian |22.9|49.8|
> |SmoothAdv|26.3 |52.5|
> |SmoothMix |16.8|52.7|
> |MACER|27.4|54.3|
> |Gaussian-CS|50.9|43.7|
> |SmoothAdv-CS|53.6 |48.8|
> |SmoothMix-CS|26.4|50.7|
> |Margin-CS|54.8|48.3|
>
>
> Thanks for the suggestion. We report the cost-sensitive robustness for both sensitive and non-sensitive examples under the CIFAR-10 S-Seed setting. As expected, there exists a trade-off in certified robustness performance between these two types of samples, which aligns with our training objective: a smaller margin threshold $\gamma_1$ is used for normal examples, while a larger margin threshold $\gamma_2$ is applied to sensitive examples.
>
> **2. The details of the attacks used in the experiments are not disclosed**
> As defined, randomized smoothing provides certified robustness against all attacks within the certified radius. For this reason, prior works on randomized smoothing—and most certification-based methods in general—do not explicitly specify a threat model or evaluate against empirical (adaptive) attacks. Following this convention, neither do we.
>
> **3. The paper does not compare its approach with the most relevant work, such as Zhang et al. (2023), nor does it include comparisons with other regular adversarial training methods.**
>
> We report the performance comparison between our method and DiffSmooth, as proposed by Zhang et al. (2023). Our Margin-CS results are presented in parentheses for direct comparison with DiffSmooth. As shown in the table, our method consistently outperforms DiffSmooth across all cost-matrix settings by a significant margin. Furthermore, our method achieves an inference time of 2.28 seconds per image, whereas DiffSmooth requires 89 seconds per image (~40 times slower), highlighting the efficiency and practicality of our approach for real-world deployment.
>
> | Method | $Acc$| $Rob_{cs}$|$Rob_{cost}$|
> |--|--|---|----|
> | S-seed| 67.12 (**67.5**) | 28.04 (**46.8**) | 4.357 (**3.04**) |
> | M-seed| 67.12 (**67.5**) | 36.45 (**54.8**) | 4.49 (**3.07**)  |
> | S-Pair| 67.12 (**67.5**) | 61.68 (**92.4**) | 0.43 (**0.05**)  |
> | M-Pair| 67.12 (**67.5**) | 33.64 (**80.4**) | 1.636 (**0.35**) |
>
> **4. The novelty of the proposed method in comparison to Zhang et al. (2023)**
> Zhang et al. 2023 enhances randomized smoothing for *general robustness* by using a diffusion model as a denoiser before passing the noisy input to the target model. In contrast, our work improves randomized smoothing for *cost-sensitive robustness* (which existing approaches cannot achieve) through a novel training paradigm. As these approaches are algorithmically distinct and directionally orthogonal, we do not see a particularly strong connection between them—beyond both falling under the broader category of randomized smoothing.
>
> **5. Specifically, the overall accuracy presented in Table 1 and Table 11 is noticeably lower than that of state-of-the-art (SOTA) adversarial training methods (refer to RobustBench).**
>
> As briefly mentioned in the above **response 2**, there is an inevitable gap between empirical robustness (i.e., potentially effective against *specific* attacks, such as adversarial training) and certified robustness (i.e., provably effective against *all* possible attacks within a bounded perturbation, e.g., randomized smoothing). Simply speaking, the noise scale incorporated in randomized smoothing framework is much larger than that in adversarial training, since empirical attacks focus on imperceptible perturbations, whereas rigorous certification requires a reliable computation of the certified radius. This difference is then reflected in the model's clean accuracy. While a deeper investigation into the gap between these two fields would be an interesting direction, it falls outside the scope of this submission.
>
> **6. The proposed method should be compared with regular adversarial training methods, as these are designed to generalize across all adversarial scenarios.**
>
> As mentioned above, the training noise used in randomized smoothing baselines is significantly larger than that used in adversarial training methods. To illustrate this discrepancy, we evaluated the certified robustness of the top two methods on RobustBench—MeanSparse and adversarial training enhanced by diffusion models—and found that their certified performance is close to zero. This highlights the fundamental mismatch between the goals of empirical adversarial training and certification-based approaches: the former aims to improve empirical robustness under small, often imperceptible perturbations, while the latter focuses on providing formal robustness guarantees under much larger perturbation regimes.

---

> > ### Comment · Reviewer_KFXn · 2025-04-02
> >
> > 1. My argument regarding the difference between adversarial training and the proposed method is related to their application, specifically in handling clean and malicious inputs during inference. The key question that needs to be addressed is: "Can adversarial training on basic Cost-Sensitive Learning defend against cost-sensitive adversarial examples? If so, why is the proposed defense necessary?" This is my primary concern regarding the motivation of this work.
> > Why can't adversarial training (e.g., [A]) be applied to basic Cost-Sensitive Learning? For instance, the simplest approach would be to introduce adversarial examples into the training dataset of basic Cost-Sensitive Learning. Please note that the authors do not need to integrate adversarial training with the proposed certified defense, but rather consider adversarial training for non-robust cost-sensitive learning. Additionally, adversarial training does not necessarily require imperceptible perturbations, so the claim in the rebuttal is incorrect.
> >
> > 2. I do not agree with the claim that a defense can protect against all attacks without evidence, especially since more advanced attacks have been proposed since 2023. Can this defense, trained with $L_2$-norm constraints, effectively defend against $L_\infty$-norm attacks or sparse attacks [B, C] with an unlimited $\epsilon$ (this is defending against unseen attacks)? Furthermore, even for the only attack considered in this paper, it is unclear which specific attack is being used. Equation 1 merely presents a general attack objective definition.
> >
> > 3. The rebuttal still does not provide clean accuracy trade-offs before and after applying the defense, despite the authors claiming in the main paper that their method mitigates this issue and arguing that adversarial training has such limitations (this is right, though for the latest adversarial training methods, this is not a significant limitation).
> >
> > 4. The rebuttal also fails to justify why recent work has not been discussed. I am concerned that this research may have been completed as early as the beginning of 2024, which would make it inappropriate for direct publication in a conference at the end of 2025 without incorporating up-to-date studies.
> >
> > [A] Zhang, H., Yu, Y., Jiao, J., Xing, E., El Ghaoui, L., & Jordan, M. (2019, May). Theoretically principled trade-off between robustness and accuracy. In ICML (pp. 7472-7482).
> >
> > [B] Su, J., Vargas, D. V., & Sakurai, K. (2019). One pixel attack for fooling deep neural networks. IEEE Transactions on Evolutionary Computation, 23(5), 828-841.
> >
> > [C] Vo, V. Q., Abbasnejad, E., & Ranasinghe, D. C. (2024). BRUSLEATTACK: A QUERY-EFFICIENT SCORE-BASED BLACK-BOX SPARSE ADVERSARIAL ATTACK. In ICLR.

---

> > > ### Author Response · Authors · 2025-04-06
> > >
> > > **For Comment 1**
> > >
> > >
> > > It is important to clarify that *certified defenses* and *empirical defenses* are two **distinct** frameworks for robustness, and **none** of the existing certified defenses considered evaluating against empirical adversarial attacks, as **certified defense provides probabilistic robustness guarantees against worst-case perturbations** [1, 2]—a guarantee that empirical methods can never offer. While SmoothAdv (Salman et al., 2019), cited in our submission, explores incorporating empirical adversarial attacks to potentially enhance certified robustness, evaluating attacks (whether with or without guarantees) falls outside the scope of this submission and, more broadly, the certified robustness literature.
> > > In addition, each input sample in certified defense is associated with a **certified radius**, which formally characterizes each sample's ability to resist perturbations. This per-sample certification is a unique property of randomized-smoothing-based approaches and is **not available in empirical adversarial training**.
> > > The *evaluation metric and pipeline* used in certified defense are also **fundamentally different** from those in empirical defense, as we have clarified in Section 6 of our paper.  While it may be a meaningful direction to explore how to bridge certified and empirical cost-sensitive learning—e.g., by aligning or unifying certified adversarial defense and empirical adversarial defense within a single framework—**this is beyond the scope of our current submission and the broader body of certified robustness work**.
> > >
> > >
> > > [1] Cohen J, Rosenfeld E, Kolter Z. Certified adversarial robustness via randomized smoothing. In *International Conference on Machine Learning*, 2019: 1310–1320.
> > > [2] Wong E, Schmidt F, Metzen J H, et al. Scaling provable adversarial defenses. *Advances in Neural Information Processing Systems*, 2018, 31.
> > >
> > >
> > > **For Comment 2**
> > >
> > >
> > > By definition, randomized smoothing-based defenses can provide **certified robustness** against all possible $\ell_2$ attacks **within a certain radius** (as already claimed in the rebuttal and the original submission), but they do **not guarantee robustness** against other types of norm-bounded attacks (e.g.,$\ell_\infty$, $\ell_1$) in their standard form (while extensions are possible such as [3],[4]), nor against unseen or unconventional adversarial strategies. Prior research [5] has also shown that adversarial training against one specific norm (such as $ \ell_\infty$) tends to **generalize poorly** to attacks in other norms (such as $ \ell_2$ or $\ell_1$).  A comprehensive investigation of this phenomenon is beyond the scope of this submission.
> > >
> > >
> > > [3]: Vorácek V, Hein M. Improving l1-certified robustness via randomized smoothing by leveraging box constraints[C]//International Conference on Machine Learning. PMLR, 2023: 35198-35222.
> > >
> > > [4]: Florian Tramer and Dan Boneh. *Adversarial Training and Robustness for Multiple Perturbations*. Advances in Neural Information Processing Systems, 2019.
> > >
> > > [5]  Yang G, Duan T, Hu J E, et al. Randomized smoothing of all shapes and sizes[C]//International conference on machine learning. PMLR, 2020: 10693-10705.
> > >
> > >
> > >
> > > **For Comment 3**
> > >
> > >
> > > The term *model accuracy* in randomized smoothing literature specifically refers to **certified clean accuracy** (of the smoothed classifier), which is fundamentally different from the *clean accuracy* typically reported in empirical adversarial training frameworks. Furthermore, all evaluation metrics used in our work are **certified robustness metrics**, computed using **Monte Carlo sampling** as part of the randomized smoothing framework. These certified metrics provide formal probabilistic guarantees under $\ell_2$-bounded adversarial perturbations, and should not be directly compared to empirical robustness metrics, which rely on adversarial attacks and do not offer worst-case guarantees.
> > >
> > > **For Comment 4**
> > >
> > >
> > > We have included the most recent works on **randomized smoothing** and **cost-sensitive learning** in our paper, supported by multiple rounds of updated literature searches. As our focus is on **certified** defenses rather than **empirical** ones, we have limited the discussion of empirical attacks and defenses, which lie outside the primary scope of our work. We would be happy to incorporate additional literature related to empirical defenses in the revision if the reviewer deems it necessary.

---

### Official Review · Reviewer_AS48 · 2025-03-11

**Overall Recommendation:** 3

**Summary:**

This paper considers certified robustness when the cost between the correct label and the incorrect one is non-uniform. Specificially, author proposed a certification method via randomized smoothing and the corresponding provable training algoriothm in this context.

**Claims And Evidence:**

The claims in this paper are convincing and supported by clear evidence.

**Essential References Not Discussed:**

N.A.

**Ethical Review Concerns:**

N.A.

**Experimental Designs Or Analyses:**

The experiments are consistent with the theoretical setup and are convincing. However, I suggest the author conduct an ablation study on hyper-parameters $\lambda_1$, $\lambda_2$, $\gamma_1$, and $\gamma_2$, showing how sensitive the performance is to these hyper-parameters and how these hyper-parameters affect the overall performance.

**Methods And Evaluation Criteria:**

The methods and evaluation criteria are mostly convincing. One concern is that the authors should report the performance variance by running the algorithm multiple times given a relatively narrow gap between the proposed method and the baselines, like some cases in Table 3.

**Other Comments Or Suggestions:**

N.A.

**Other Strengths And Weaknesses:**

The strengths and weaknesses are demonstrated in the sections above. Overall, I think it is a good paper. I welcome the authors to address my concerns in the rebuttal and will re-evaluate the manuscript during the discussion period.

**Questions For Authors:**

The questions are summarized as follows:

1. The authors should report the performance variance by running the algorithm multiple times given a relatively narrow gap between the proposed method and the baselines, like some cases in Table 3.

2. I suggest the author conduct an ablation study on hyper-parameters $\lambda_1$, $\lambda_2$, $\gamma_1$, and $\gamma_2$, showing how sensitive the performance is to these hyper-parameters and how these hyper-parameters affect the overall performance.

**Relation To Broader Scientific Literature:**

The provable robustness is a crucial problem in situations where the tolerance of mistakes is very low. The motivation and the applicable situations of the method make sense. I believe this paper can contribute and raise some interest in the adversarial machine learning community.

**Theoretical Claims:**

The theoretical claims are convincing, while the proof technique is based on Cohen et al 2019.

---

> ### Author Rebuttal · Authors · 2025-03-31
>
> **1. The authors should report the performance variance**
>
> We conduct three independent runs of the Imagenette S-Seed setting and report the corresponding standard deviations. The experimental procedure comprises two distinct stages: the training phase and the certification (evaluation) phase. The majority of variance arises during training, while variance during certification is negligible due to extensive Monte Carlo sampling and majority voting, which together ensure stable and consistent results. As shown, running all the baselines and our method multiple times does not affect the claims nor findings. We will include performance variance details to further strengthen the manuscript.
>
> | Methods       | $Acc$ (± std)   | $Rob_{cs}$ (± std)   | $Rob_{cost}$ (± std)   |
> |---------------|---------------|---------------|----------------|
> | Gaussian      | 80.3 (0.22)    | 64.6 (0.32)    | 3.67 (0.006)   |
> | SmoothAdv     | 80.2 (0.17)    | 64.3 (0.25)    | 3.91 (0.007)   |
> | SmoothMix     | 80.6 (0.19)    | 59.5 (0.50)    | 2.93 (0.032)   |
> | MACER         | 78.2 (0.10)    | 63.8 (0.16)    | 2.46 (0.004)   |
> | Gaussian-CS   | 74.6 (0.09)    | 73.3 (0.24)    | 1.67 (0.009)   |
> | SmoothAdv-CS  | 77.6 (0.14)    | 66.6 (0.31)    | 3.82 (0.016)   |
> | SmoothMix-CS  | 76.1 (0.09)    | 68.9 (0.48)    | 2.24 (0.042)   |
> | Margin-CS     | 79.6 (0.01)    | 81.1 (0.15)    | 1.35 (0.008)   |
>
> **2. Ablation study on hyper-parameters**
>
> There are two stages in the hyperparameter tuning process. In the first stage, $\lambda_1$ and $\lambda_2$ control the overall trade-off between certified accuracy and cost-sensitive robustness. As expected, increasing $\lambda_1$ and $\lambda_2$ leads to a decrease in overall accuracy but an improvement in cost-sensitive robustness. We set $\lambda_1 = \lambda_2 = 3$ to achieve a favorable balance between the two objectives.
>
> In the second stage, we fix $\lambda_1$ and $\lambda_2$ and tune $\gamma_1$ and $\gamma_2$. Here, $\gamma_1$ determines the margin threshold for selecting normal samples, while $\gamma_2$ controls the threshold for sensitive samples. We observe that increasing $\gamma_2$ enhances cost-sensitive performance, whereas increasing $\gamma_1$ improves overall accuracy, but will decrease cost-sensitive performance. A grid search reveals that the combination $(\gamma_1, \gamma_2) = (4, 16)$ yields satisfactory results.
>
> We will include these details in the revised manuscript's Appendix.
>
> | $\lambda_1$ |$\lambda_2$|$Acc$| $Rob_{cs}$|$Rob_{cost}$|
> |-|-|-|-|-|
> | 1 | 1 | 0.690 | 0.22| 5.150 |
> | 2 | 2 | 0.682| 0.435  | 3.619 |
> | 3 | 3 | 0.667 | 0.510|3.407 |
> | 4| 4 | 0.631| 0.732 | 1.597|
> | 5 | 5 | 0.603 | 0.762 | 1.367|
> | 6 | 6 | 0.578 | 0.811| 1.046|
>
> | γ1  |$Acc$(γ2=8) | $Rob_{cs}$(γ2=8) | $Rob_{cost}$ (γ2=8) | $Acc$ (γ2=10) | $Rob_{cs}$ (γ2=10) |$Rob_{cost}$ (γ2=10) | $Acc$(γ2=12) | $Rob_{cs}$(γ2=12)|$Rob_{cost}$(γ2=12) | Acc (γ2=16) |$Rob_{cs}$(γ2=16) | $Rob_{cost}$(γ2=16)|
> |-----|-- |-- |-- |-- |-- |--|--|--|--|--|-|--|
> |  2| 65.4| 63.3| 2.617| 63.4 | 68.7| 2.488| 63.7| 69.1| 2.484|63.0|70.5   | 2.505|
> |  4| 68.2| 49.7| 3.558| 67.9| 52.6| 3.477| 67.7| 54.3| 3.444| 67.5 | 54.8| 3.040|
> |  6| 67.3| 39.6| 3.878| 66.0| 49.3| 3.546|65.5|54.4| 3.362 | 64.9 | 55.2 | 3.231 |
> | 8 | 66.0 | 33.8  |4.390 | 65.0 | 43.2 | 4.137| 64.1 | 47.4 | 3.921 | 64.5 | 46.3 |3.768 |

---

### Official Review · Reviewer_789m · 2025-03-18

**Overall Recommendation:** 5

**Summary:**

The paper introduces a novel framework for adversarial robustness that incorporates cost-sensitive learning using randomized smoothing. Unlike existing defenses that assume uniform misclassification costs, this method optimizes robustness with a cost matrix that accounts for real-world risk variations (e.g., misclassifying malignant tumors as benign is costlier than the reverse). The main contributions include:

- Cost-Sensitive Certified Radius: A new metric extending the standard certified radius to account for cost-sensitive adversarial robustness.
- Certification Algorithm: A Monte Carlo-based method to estimate cost-sensitive robustness, ensuring statistically rigorous bounds.
- Robust Training Method: A margin-based loss function that maximizes cost-sensitive certified robustness while maintaining accuracy.
- Experimental Validation: The approach outperforms baselines such as standard randomized smoothing, SmoothAdv, and MACER on CIFAR-10, Imagenette, ImageNet, and the medical HAM10k dataset.

The results indicate that the method significantly enhances robustness, particularly in cost-sensitive scenarios, making it relevant for safety-critical applications.

**Claims And Evidence:**

The paper's main claims are:

1. Cost-sensitive certified radius provides better robustness guarantees (Theorem 4.2).  Supported by theoretical proof showing it generalizes the standard certified radius.
2. Monte Carlo-based certification is statistically valid (Theorem 4.4). Proof provided using a union bound argument.
3. Proposed training method (Margin-CS) improves cost-sensitive robustness without degrading accuracy. Extensive experimental results demonstrate a ~20% improvement over baseline methods.
4. Scalability to high-dimensional models (e.g., ImageNet). Empirical evidence shows that the approach works on large datasets where previous methods struggle.

All claims are well-supported, with both theoretical and empirical validation.

**Essential References Not Discussed:**

NA

**Experimental Designs Or Analyses:**

- Certifications for cost-sensitive robustness are evaluated on diverse datasets and settings.
- Robustness is tested across multiple perturbation budgets ($\epsilon$-values).
- Results on HAM10k confirm the method's relevance to medical AI.

Weakness
- No ablation study on the effectiveness of different training components (e.g., margin-based loss vs. alternative approaches).
- Robustness to $l2$ perturbations (e.g., adversarial patches, feature space attacks) is not explored.

**Methods And Evaluation Criteria:**

- The benchmark datasets (CIFAR-10, Imagenette, ImageNet, HAM10k) are appropriate, covering both general vision tasks and real-world cost-sensitive applications (e.g., medical imaging).
- The metrics used (Certified Robust Cost, Certified Cost-Sensitive Robustness, Certified Accuracy) effectively measure both overall robustness and cost-sensitive robustness.
- The comparison to baselines is thorough, including standard randomized smoothing and adversarial training methods.

**Other Comments Or Suggestions:**

- Sensitivity Analysis: Adding a hyperparameter sensitivity analysis ($\sigma$, $\delta$, $\gamma$) would improve the practical usability of the method.
- Adaptive Adversarial Attacks: Evaluating the approach against adaptive attacks (e.g., AutoAttack, patch-based attacks) would strengthen the robustness claims.
- Ablation Studies: Analyzing the contributions of individual training components (margin-based loss vs. standard cost-sensitive learning) would clarify which aspects drive performance gains.

**Other Strengths And Weaknesses:**

Strengths
- Strong Theoretical Guarantees: Provides mathematical proofs for all major claims.
- Scalable to Large Models: Works on ImageNet and HAM10k, where previous cost-sensitive methods struggled.
- Real-World Applicability: Demonstrates importance in medical AI.
- General Framework: Extends randomized smoothing to cost-sensitive settings in a principled manner.

Weaknesses
- No Adaptive Attack Evaluation: The method is only tested under $l2$-norm perturbations.
- Lack of Hyperparameter Sensitivity Analysis: Does not explore robustness to different $\sigma$, $\delta$ values.
- Limited Discussion on Real-World Deployment: Practical constraints (e.g., computational cost, real-time processing) are not discussed.

**Questions For Authors:**

1. How does the method perform under adaptive attacks? Randomized smoothing is not foolproof against strong adaptive attacks (e.g., query-based attacks). Have you tested against adaptive adversarial strategies, and if so, how does the method hold up?
What is the computational overhead of certification?

2. Monte Carlo-based certification can be computationally expensive. How does the certification time scale with input dimensions and the number of classes? Can the method be extended to other robustness frameworks?

**Relation To Broader Scientific Literature:**

The paper is highly relevant to ongoing research in:

- Certified Adversarial Robustness: Extends works like Cohen et al. (2019) on randomized smoothing.
- Cost-Sensitive Learning: Builds on prior methods like Zhang & Evans (2019) but improves scalability and certification guarantees.
- Medical AI and Risk-Aware ML: Addresses real-world concerns in safety-critical AI (e.g., healthcare, autonomous systems).

The work bridges the gap between adversarial robustness and cost-sensitive classification, making it an important contribution.

**Theoretical Claims:**

I checked the following theoretical claims:

- Theorem 4.2 (Cost-sensitive certified radius is always greater than or equal to the standard certified radius). Proof follows from standard randomized smoothing principles and the monotonicity of $\phi^{-1}$. Mathematically sound and intuitive.

- Theorem 4.4 (Certified robustness estimate is statistically valid). Proof relies on union bounds and confidence interval estimation. The logic appears correct, but empirical verification (e.g., checking Monte Carlo estimates converge) would strengthen confidence.

---

> ### Author Rebuttal · Authors · 2025-03-31
>
> **1. Theorem 4.4 (Certified robustness estimate is statistically valid). The logic appears correct, but empirical verification (e.g., checking Monte Carlo estimates converge) would strengthen confidence.**
>
> We conduct the certification process using varying numbers of Monte Carlo samples, where N denotes the number of samples and r represents the certified radius returned by Algorithm 1. The results for the S-Seed setting are shown in the table below. We observe that as N increases, the certified radius gradually converges. Specifically, once N exceeds 50,000, the certified radius stabilizes and exhibits minimal fluctuation.
> | N      | 100   | 500   | 1000  | 10000 | 50000 | 60000 | 70000 | 80000 | 90000 | 100000 |
> |--------|-------|-------|-------|-------|--------|--------|--------|--------|--------|---------|
> | r   0.4378| 0.6869| 0.7663| 0.9550| 1.0612 | 1.052  | 1.059  | 1.065  | 1.069  | 1.072   |
>
>
> **2. No Adaptive Attack Evaluation: The method is only tested under L2 norm perturbations.**
>
> As defined, randomized smoothing provides certified robustness against all attacks—including adaptive ones—within the certified radius. For this reason, prior works on randomized smoothing (and most certification-based methods in general) do not evaluate against adaptive attacks. While we agree that exploring the gap between certified robustness and empirical robustness (i.e., performance under actual attacks) is an interesting direction, it falls outside the scope of this submission.
>
> Although randomized smoothing is formally defined for L2 norm perturbations, prior work has shown that it can be effectively extended to other norms [1,2]. As this generalization is well-established and independent of our main contributions, we do not focus on it here, but we are happy to provide relevant results upon request.
>
> [1] Yang, Greg, et al., "Randomized Smoothing of All Shapes and Sizes", ICML 2020
> [2] Vorácek, V., & Hein, M.,  "Improving L1-Certified Robustness via Randomized Smoothing by Leveraging
> Box Constraints", ICML 2023
>
> **3. Limited Discussion on Real-World Deployment: Practical constraints (e.g., computational cost, real-time processing) are not discussed.**
>
> As suggested, we report the training and inference time evaluated on a single A100 GPU (40 GB Memory). During training, we employ a margin-based loss to optimize the cost-sensitive certified radius, which necessitates a substantial amount of Gaussian sampling (e.g. 16) for accurate radius estimation, as well as sufficient epochs to ensure convergence. This results in a slight increase in the training time. During inference, they all share the same certification procedure, so their certification time is nearly identical.
>
> | Method| Gaussian | SmoothAdv | SmoothMix | MACER | Gaussian-CS | SmoothAdv-CS | SmoothMix-CS | Margin-CS |
> |----------------|----------|-----------|-----------|-------|-------------|---------------|--------------|-----------|
> | Training Time  |0.2h |11.53h| 2.68h|13.5h|0.2h|11.53h| 2.69h|13.4h |
> | Inference Time | 2.38s |2.27s| 2.28s | 2.25s | 2.39s | 2.27s | 2.28s        | 2.28s|
>
>
> **4. What is the computational overhead of certification? How does Monte Carlo certification time scale with input dimensions and the number of classes? Can the method be extended to other robustness frameworks?**
>
> The overall certification cost is $O(N\times T_{fwd})+O(C)$, where $N$ is the Gaussian sampling number and $T_{fwd}$ is the cost per forward pass (depends on model size and input dim), $C$ is the number of classes.
> $O(C)$ is due to post-processing steps after the Monte Carlo sampling. This includes operations like finding the top classes, computing confidence bounds, and estimating per-class certified radii. While this cost is small compared to the sampling phase, it scales linearly with the number of classes
> $C$. Our certification algorithm is compatible with any robustness framework. It is independent of the training procedure and can also be integrated into diffusion-based certification methods (as shown in the reponse to **Reviewer KFXn**), The values in parentheses correspond to our method.
>
> | Setting| Overall Acc| $Rob_{cs}$|$Rob_{cost}$|
> |--|--|---|----|
> | S-seed| 67.12 (**67.5**) | 28.04 (**46.8**) | 4.357 (**3.04**) |
> | M-seed| 67.12 (**67.5**) | 36.45 (**54.8**) | 4.49 (**3.07**)  |
> | S-Pair| 67.12 (**67.5**) | 61.68 (**92.4**) | 0.43 (**0.05**)  |
> | M-Pair| 67.12 (**67.5**) | 33.64 (**80.4**) | 1.636 (**0.35**) |
>
> **5. Ablation Study & Sensitivity Analysis: Adding a hyperparameter sensitivity analysis would improve the practical usability of the method.**
>
> Please refer to our **Response 2 to Reviewer AS48** for the hyperparameter sensitivity analysis. The experimental results show that the parameter $\lambda$, $\gamma$ controls the trade-off between overall accuracy and cost-sensitive performance. We will include more details on the ablation study and sensitivity analysis in the Appendix of the revised manuscript.

---

### Decision · Program_Chairs · 2025-05-01

**Decision:**

Accept (poster)

**Comment:**

The paper introduces a framework for adversarial robustness that integrates cost-sensitive learning with randomized smoothing. Some reviewers noted that the method significantly improves robustness, particularly in cost-sensitive settings, making it relevant for safety critical applications. There were suggestions to include ablation studies to further support the results. Several comments pointed out the need for a stronger discussion of related work and a clearer comparison with prior approaches.